# A nuclear protein quality control system for elimination of nucleolus-related inclusions

Lorène Brunello [1], Jolanta Polanowska [1], Léo Le Tareau [1], Chantal Maghames[1], Virginie Georget[1,2], Charlotte Guette [3], Karima Chaoui[4], Stéphanie Balor[3], Marie-Françoise O'Donohue [3], Marie-Pierre Bousquet[4], Pierre-Emmanuel Gleizes[3] & Dimitris P Xirodimas [1]✉

## Abstract

**The identification of pathways that control elimination of protein inclusions is essential to understand the cellular response to proteotoxicity, particularly in the nuclear compartment, for which our knowledge is limited. We report that stress-induced nuclear inclusions related to the nucleolus are eliminated upon stress alleviation during the recovery period. This process is independent of autophagy/lysosome and CRM1-mediated nuclear export pathways, but strictly depends on the ubiquitin-activating E1 enzyme, UBA1, and on nuclear proteasomes that are recruited into the formed inclusions. UBA1 activity is essential only for the recovery process but dispensable for nuclear inclusion formation. Furthermore, the E3 ligase HUWE1 and HSP70 are components of the ubiquitin/chaperone systems that promote inclusion elimination. The recovery process also requires RNA Pol I-dependent production of the lncRNA IGS$_{42}$ during stress. IGS$_{42}$ localises within the formed inclusions and promotes their elimination by preserving the mobility of resident proteins. These findings reveal a protein quality control system that operates within the nucleus for the elimination of stress-induced nucleolus-related inclusions.**

**Keywords** Nucleolus; Proteotoxic Stress; Ubiquitin/NEDD8; Proteasome; IGS ncRNAs
**Subject Categories** Post-translational Modifications & Proteolysis; Translation & Protein Quality

## Introduction

Cells are constantly exposed to environmental stresses, including exposure to high temperatures, oxidative stress, or acidosis that cause protein misfolding. The protein quality control (PQC) system ensures the detection and repair of misfolded proteins or the elimination of terminally damaged proteins through the ubiquitin-proteasome (UPS) and autophagy systems (Amm et al, 2014; Sontag et al, 2014, 2017;

Kandel et al, 2023). Misfolded proteins have the propensity to form higher-order complex intermediates, usually containing RNA (Fassler et al, 2021). In particular, the role of long noncoding RNAs (lncRNA) in the formation of such complexes upon stress conditions has received wide attention. This is due to the ability of lncRNAs to function as nucleation factors that enables the formation of dynamic and reversible RNA-protein assemblies (Louka et al, 2020; Onoguchi-Mizutani and Akimitsu, 2022). However, depending on the length or strength of the applied stress conditions and the presence of mutations in resident proteins, these complexes can often convert into insoluble aggregates, which are regarded as hallmarks of neurodegenerative diseases and ageing (Amm et al, 2014; Yoo et al, 2022; Sontag et al, 2017; Hipp et al, 2014; Vilchez et al, 2014). The UPS is established as a pathway for the elimination of soluble terminally misfolded proteins in order to prevent aggregation, whereas autophagy eliminates protein inclusions/aggregates through a selective process that involves autophagy cargo receptors such as p62/SQSTM1 (Dantuma and Bott, 2014; Gubas and Dikic, 2022).

It is now evident that compartmentalization of protein inclusions is a critical and highly conserved element of the cellular response to stress (Escusa-Toret et al, 2013; Specht et al, 2011; Hill et al, 2017). For example, in *Saccharomyces cerevisiae*, stress causes the generation of protein inclusions in distinct cytoplasmic compartments, including the juxtanuclear quality control compartment (JUNQ), the perivacuolar insoluble protein deposit (IPOD), or the ER-associated compartments (ERAC) (Chen et al, 2011; Kaganovich et al, 2008; Huyer et al, 2004). The generation of JUNQ is characterized by the accumulation of ubiquitinated misfolded proteins that retain mobility, whereas IPOD contains immobile terminally aggregated proteins (Chen et al, 2011; Kaganovich et al, 2008). Additionally, in mammalian cells, distinct perinuclear deposits induced upon heat shock or proteasome inhibition, known as aggresomes and aggresome-like inclusions, have been extensively characterized. These inclusions are formed at the microtubule organizing center within the intermediate filaments such as vimentin or keratin (García-Mata et al, 1999; Johnston et al, 1998).

While the above described-processes are well characterized for the cytoplasmic PQC system, our knowledge on how the nucleus deals with misfolded proteins and protein inclusions is limited, especially in mammalian cells (Franić et al, 2021; Enam et al, 2018). A well-described

[1]CRBM, Univ. Montpellier, CNRS, Montpellier, France. [2]MRI, BioCampus, Univ. Montpellier, CNRS, INSERM, Montpellier, France. [3]Molecular, Cellular and Developmental biology department (MCD), Centre de Biologie Intégrative (CBI), University of Toulouse, CNRS, UT3, Toulouse, France. [4]Institut de Pharmacologie et de Biologie Structurale, CNRS, Université Paul Sabatier (UPS), Université de Toulouse, Toulouse 31000, France. ✉E-mail: dimitris.xirodimas@crbm.cnrs.fr

type of nuclear inclusions induced by heat shock (HS), acidosis, or proteasome inhibition is related to the nucleolus, the center of ribosome biogenesis (Mélèse and Xue, 1995; Hori et al, 2023; Batnasan et al, 2022). Upon stress, the nucleolus undergoes a dramatic reorganization, where a fraction of the nucleolar proteome that includes mainly ribosomal proteins (RPs) is found in nuclear deposits that are either within or adjacent to the nucleolar structure, described as nucleolar aggresomes (Latonen, 2019; Latonen et al, 2011; Maghames et al, 2018). Similar nuclear inclusions have been described in disease model systems such as Huntington's disease and spinocerebellar ataxias (Davies et al, 1997) and may be also related to nuclear deposits observed in *S. cerevisiae*, within the so-called intranuclear quality control compartment (INQ) (Miller et al, 2015; Sontag et al, 2023; Kumar et al, 2022; Gallina et al, 2015). This response is related to the natural ability of the nucleolus to act as a detention center for several cellular factors, where upon stimulus, including cell cycle perturbations, can reversibly transit from a mobile to an immobile state through an amyloidogenesis process (Boulon et al, 2010; Frottin et al, 2019; Wang et al, 2018, 2019; Chuang et al, 2018). Interestingly, these stress-induced inclusions are "decorated" with ubiquitin and ubiquitin-like molecules (Ubls) including SUMO-2 and NEDD8, presumably as markers of protein misfolding (Latonen et al, 2011; Maghames et al, 2018; Lobato-Gil et al, 2020; Maynard et al, 2009). The E3-ligase HUWE1 was identified as the main enzyme responsible for the modification of nucleolus-related protein inclusions with ubiquitin and NEDD8 (Maghames et al, 2018). The biological outcome of the formation of these inclusions is, however, not fully understood. It is suggested that they are involved in the DNA replication stress response, to specifically protect the nuclear proteasome from collapsing during stress conditions due to the accumulation of misfolded proteins and/or to act as a safeguard mechanism for epigenetic regulators during stress (Mediani et al, 2019; Frottin et al, 2019; Maghames et al, 2018; Miller et al, 2015; Azkanaz et al, 2019). It is also unclear how the nucleus copes with such inclusions during the recovery period and particularly what is the role of the ubiquitin system in this process (Shibata and Morimoto, 2014). The redistribution of nuclear inclusions in the cytoplasm during mitosis and their elimination by the autophagy system in the subsequent cell cycle has been proposed as a recovery mechanism (Nyström and Liu, 2014; Rujano et al, 2006). On the other hand, studies on artificial model substrates and pathological Huntingtin-related aggregates, indicated that nuclear aggregates can be eliminated in a proteasome-dependent but autophagy-independent manner. This process also requires the HSP70 chaperone that binds to misfolded client proteins (Hjerpe et al, 2016). However, it is still unclear whether a PQC system operates within the nucleus for the elimination of stress-induced aggregates.

Here, we report that nucleolus-related inclusions are transient and are eliminated upon stress alleviation. The recovery process is independent of the CRM1-mediated nuclear export and the autophagy/lysosome pathway, but strictly depends on the activity of nuclear proteasomes that are recruited within the inclusions upon stress. The ubiquitin pathway and the HUWE1 E3-ligase are essential for the elimination but not for the generation of the inclusions. We also found that the elimination process requires the RNA Pol I-dependent production of lncRNAs ($IGS_{42}$) within the intergenic spacer region of rDNA during stress. The induced lncRNA localizes within the formed inclusions and maintains the mobility of resident proteins as a critical step for the recovery process. Collectively, the study defines components of a PQC system that operates within the nucleus for the elimination of nucleolus-related inclusions.

# Results

## Severe stress causes the reorganization of the nucleolus

Stress induced by HS or partial proteasome inhibition using MG132, causes the generation of nucleolus-related inclusions (Latonen, 2019). Consistent with this notion, by monitoring the localization of fibrillarin and RPL11 as nucleolar markers that localize within distinct nucleolar compartments, we found that both HS and MG132 cause nucleolar reorganization but to different extents. During HS, RPL11 accumulates in the core of the nucleolus while fibrillarin localizes at the periphery in a structure that looks similar to "nucleolar caps" (Shav-Tal et al, 2005). As evidence of the stress response and generation of misfolded proteins, the nucleolus is also stained with ubiquitin (Fig. 1A,B; Appendix Fig. S1). This is consistent with the modification of nucleolar proteins such as RPs with ubiquitin but also with hybrid chains composed of ubiquitin and Ubls, including NEDD8 and SUMO-2 (Maghames et al, 2018; Lobato-Gil et al, 2020; El Motiam et al, 2019; Martín-Villanueva et al, 2021). However, a more dramatic reorganization was observed upon MG132, where RPL11 was clearly separated from fibrillarin (Fig. 1A; Appendix Fig. S1). Intriguingly, under these conditions, ubiquitin stained only the RPL11-formed inclusions, generating similar to previously reported "ring-like" structures or nucleolar aggresomes (Latonen et al, 2011; Maghames et al, 2018; Gallardo et al, 2020) (Fig. 1C). This also confirms the specificity of the ubiquitin system towards misfolded protein inclusions. We found that additional RPs, such as RPS7 and RPL7, also accumulate in ubiquitin-stained inclusions upon MG132 treatment, similarly to RPL11, indicating that inclusion formation upon MG132-induced stress is a common characteristic of RPs (Appendix Fig. S2A).

To characterize with higher resolution the stress-induced nucleolar reorganization, we applied expansion microscopy (Asano et al, 2018). The technique is based on the use of polymers that allow the physical expansion of biological structures while retaining the original details that permits high-resolution imaging (<50 nm) using conventional microscopes. In control cells, RPL11 is localized within the granular component of the nucleolus that surrounds fibrillarin (dense fibrillar component marker) and RPA194 (catalytic subunit of RNA Polymerase I, fibrillar center marker) (Fig. 1D,E). Upon MG132 stress, the nucleolar architecture is dramatically altered, with the exclusion of RPL11 from the two nucleolar components that contain fibrillarin and RPA194 (Fig. 1D,E). Live-imaging microscopy during stress, confirmed the progressive separation of RPL11 from other nucleolar components (nucleolin) (Fig. 1F; Movie EV1), which was completed within 7 to 10 h of stress.

To further characterize in more detail the MG132-induced nucleolar reorganization and the generation of RPs inclusions, we performed CLEM (correlative-light electron microscopy) that allows to obtain fluorescent images of RPL11 and high-resolution structural information of the nucleolus. CLEM images upon MG132 treatment showed the fragmentation of the nucleolus and the generation of inclusions containing RPL11 (Fig. 1G,H). Interestingly, similarly to what is observed with nuclear inclusions formed with mutant Huntingtin HTT, the architecture of the nucleolus-related inclusions is quite dense, homogenous and do not form a core and shell (Riguet et al, 2021; Nazarov et al, 2022). Moreover, we observed that the ultrastructure of the MG132-treated nucleolus harbors many large vacuoles (regions similar to

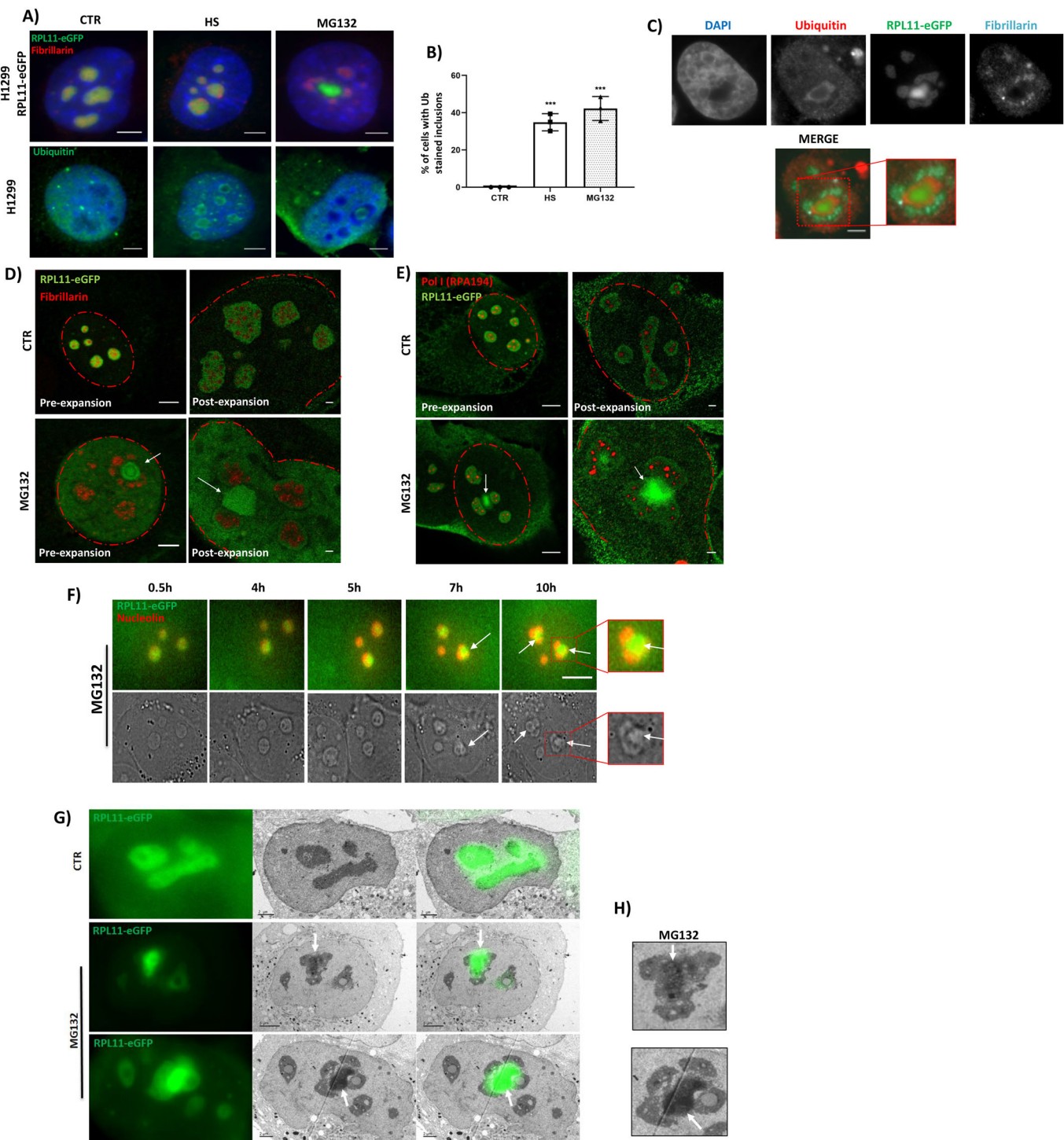

**Figure 1. Reorganisation of the nucleolus upon stress and the generation of nucleolus-related inclusions.**

(**A**) H1299 cells stably expressing RPL11-eGFP (top) and parental H1299 cells (bottom) were stressed with HS (2 h, 43 °C) or MG132 (5 μM, 15 h) before being stained for fibrillarin (red, upper part) or ubiquitin (green, lower part). Nuclei are stained with DAPI (blue). Scale bar 5 μm. (**B**) Quantitation of the experiment in A showing % of cells with nuclear inclusions (ubiquitin nuclear "ring-like" staining). Each dot represents an independent experiment ($n = 3$). *** $P \leq 0.001$ by Student $t$-test. (**C**) MG132-treated RPL11-eGFP cells are stained with ubiquitin (red), fibrillarin (cyan), and DAPI (blue). (**D**, **E**) Expansion microscopy (ExpM) imaging of H1299 RPL11-eGFP cells in control or MG132-treated conditions. The left panels correspond to confocal images from non-expanded cells (pre-expansion), while the right panels represent cells following the expansion protocol. Fibrillarin (red) (**D**) is used as a marker of the dense fibrillar component, whereas RPA194 (RNA Pol I) (**E**) as a marker for the fibrillar center. Arrows indicate the position of inclusions. (**F**) RPL11-eGFP H1299 cells were transfected with iRFP-nucleolin prior to MG132 (5 μM) treatment. Images at different time points from live imaging. Arrows indicate the position of inclusions. Scale bar 10 μm. (**G**) CLEM images of the nucleolus untreated-DMSO (CTR) and MG132-treated cells. On the right panel, merge the fluorescence image with an electron microscopy image. Localization of RPL11 inclusions is marked with a white arrow. Scale bar 2 μm. (**H**) Enlarged images of the experiment performed in (**G**). Source data are available online for this figure.

the nucleoplasm in appearance in EM sections) compared to the nucleolus in control cells (Fig. 1G,H). Collectively, the data indicate that severe stress induced upon prolonged proteasome inhibition induces a dramatic nucleolar reorganization that leads to the generation of inclusions containing RPL11.

## RPs are immobilized upon stress

The nucleolus represents one of the best-characterized membrane-less organelles governed by principles of liquid-liquid phase separation (LLPS) (Feric et al, 2016; Latonen, 2019; Wang et al, 2019). A key biophysical property of LLPS structures is the mobility of proteins within the phase. Fluorescence Recovery After Photobleaching (FRAP) within the nucleolus during the stress period (HS) showed that the RPL11-eGFP mobile fraction is dramatically reduced (Fig. 2A,B), indicating the transition of RPs from a LLPS to the so-called liquid-solid phase separation (LSPS), also observed during formation of nucleolus-related amyloid bodies (A-bodies) (Wang et al, 2018). Consistent with previous studies, 1 h of HS is sufficient to immobilize RPL11 within the nucleolus, whereas for MG132, a decrease in the RPL11 mobile fraction is observed upon 7 h of treatment, when RPL11 inclusions begin to form (Frottin et al, 2019) (Fig. 2C–E). In contrast to control conditions where RPL11 is homogenously localized within the nucleolus, upon prolonged treatment with MG132 (15 h), RPL11 displays two different localization profiles: Within the re-organized nucleolus area where fibrillarin and RNA Pol I are also found (Fig. 1D,E) (state A) and within the induced inclusions, characterized by high fluorescence intensity (state B) (Fig. 2C). This now allows us to assess the biophysical properties of a model substrate protein within the two nucleolar states induced upon stress. FRAP analysis on RPL11-eGFP in MG132-treated cells, either in state A or in state B, shows that while a high mobile fraction of RPL11 is retained within the nucleolus (state A), RPL11 is totally immobile within the induced inclusions (state B, Fig. 2D,E). This now provides biophysical evidence that the reorganization of the nucleolus upon stress results in the generation of immobile nuclear inclusions that most likely contain non-functional damaged RPs. The data are also consistent with the idea that the nucleolus is a compartment with distinct coexisting separation phases (Feric et al, 2016).

## The generation of HS and MG132-induced nucleolus-related inclusions is a reversible process

The generation of stress-induced nucleolus-related inclusions is well documented, and reversibility has been reported for nuclear inclusions induced by HS or acidosis (Audas et al, 2016; Wallace et al, 2015; Saad et al, 2017; Nollen et al, 2001; Ali et al, 2023). However, whether MG132-induced nuclear inclusions are eliminated during the recovery period upon stress alleviation is unknown. Based on immunofluorescence and western blot analysis of ubiquitin and NEDD8, we found that MG132 and HS-induced nuclear inclusions are progressively eliminated during recovery (Fig. 3A–C; Appendix Fig. S2B,C). We also monitored the fate of RPL11 during the recovery period as a model substrate for nuclear inclusion formation. Similarly, to what is observed with ubiquitin, RPL11 is progressively eliminated from the MG132 and HS-induced inclusions and within 24 h RPL11 fully recovers its

localization relative to fibrillarin, indicative of the re-formation of active nucleoli (Fig. 3D,E; Appendix Fig. S2D). To exclude the possibility that the apparent inclusion elimination is due to cell death, we monitored RPL11-eGFP cells by live imaging. As illustrated in Fig. 3F; Movie EV2, RPL11 inclusions are eliminated in individual cells during recovery.

While the data strongly indicate the presence of a recovery process, the fate of the proteins within the formed inclusions during the recovery period remains unclear. One possibility is that proteins are eliminated/degraded during recovery, or instead, they refold and reverse to a soluble state within a newly formed nucleolus. To distinguish between these possibilities, we employed a photoconversion approach to specifically monitor by live imaging the fate of a protein within the inclusions during the recovery period. For this, we used RPL11 as a model substrate fused to mEos2. Upon irradiation, the green fluorescence of mEos2 switches to red, allowing monitoring the fate of a specific pool of the protein interest (McKinney et al, 2009). We confirmed that RPL11-mEos2 localizes to the nucleolus, and upon MG132-induced stress, a fraction is re-localized within the formed inclusions, similarly to what we observe with RPL11-eGFP (Appendix Fig. S2E). We applied the following criteria to ensure that we photoconvert RPL11-mEos2 within the stress-induced inclusions: (1) A key characteristic of RPL11 within inclusions is the emission of bright intense green fluorescence compared to other parts of the nucleolus and (2) Photoconversion of RPL11-mEos2 of an unstressed nucleolus results in almost spontaneous appearance of red RPL11 into neighboring nucleoli, indicative of the dynamic exchange of nucleolar proteins (Appendix Fig. S2F; Movie EV3). In direct contrast, photoconverted RPL11 within the stress-induced inclusions remains intact with no diffusion to neighboring nucleoli (Fig. 3G; Movie EV4). Using these criteria, we followed the fate of photoconverted RPL11 within the stress-induced inclusions for 10 h during recovery. In an average of ~70% of monitored inclusions (~100 inclusions in a total of four independent experiments), the photoconverted RPL11 is eliminated, with no indication of re-assembly into a functional nucleolus (Fig. 3G; Appendix Fig. S2G; Movie EV4). Consistent with the immuno-fluorescence analysis, a fraction of cells (~30%) fail to process such inclusions during the employed recovery period. Additionally, in control, unstressed cells, emission from the photoconverted RPL11 within the nucleolus remained intact during the 10 h image acquisition period (Movie EV3), excluding the possibility that the decrease in the emission of photoconverted RPL11 observed in ~70% of stressed cells is due to instability of the fluorescent tag. Additionally, under the same conditions, no changes in gene expression for tested RPs (RPL7, RPL11, RPS7) were detected, suggesting that the observed decrease in photoconverted RPL11 fluorescence is due to post-transcriptional mechanisms (Appendix Fig. S2H). Collectively, the data show that the generation of nucleolus-related inclusions is a reversible process, where the majority of inclusions are eliminated during recovery.

Our study has focused on monitoring by imaging the accumulation of some RPs within stress-induced nuclear inclusions. To obtain a global view of the collective behavior of proteins within the formed inclusions, we followed a SILAC-based quantitative mass-spectrometry approach. We adapted protocols for the isolation of N-lauroylsarcosine insoluble Alzheimer disease aggregates and HS-induced inclusions to enrich for nuclear

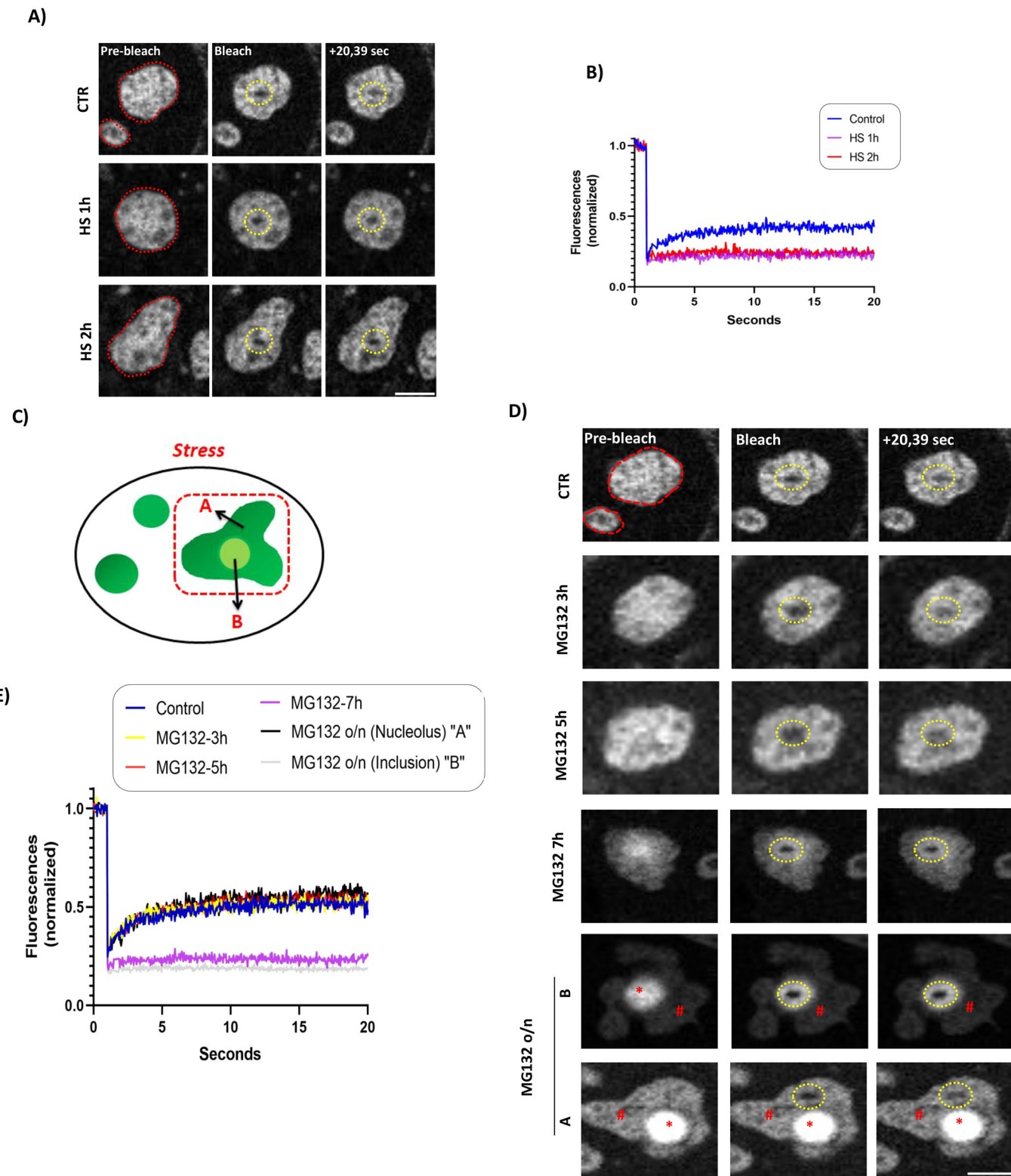

aggregates (Methods) (Diner et al, 2017; Maghames et al, 2018). We first determined the proteins that accumulate in inclusions upon MG132 treatment. We labeled cells either with light (Lys0/Arg0) or heavy (Lys8/Arg10) amino acids and were either unstressed (light) or stressed (heavy) with MG132 (15 h). An equal number of cells were mixed before inclusion isolation (Fig. 3H, Methods). Quantitation of data from three independent experiments, identified 2440 proteins enriched in MG132-induced inclusions (Fig. 3I; Dataset EV1). Gene ontology analysis of the proteomics data revealed that components of the cell cycle, nucleocytoplasmic

**Figure 2.   Heat shock and MG132 reduce the mobility of RPL11.**

(A) H1299 cells stably expressing RPL11-eGFP were left unstressed or heat shocked for 1 or 2 h prior to FRAP analysis. The red circle represents the nucleoli. Cells were bleached in the indicative yellow (circle) regions and allowed to recover. (B) Fluorescence recovery kinetics of the bleached regions in (A) were quantified and presented as the median relative intensity of 15 different cells. Scale bar 5 µm. (C) Schematic representation for the reorganization of the nucleolus under proteotoxic stress, where mobile RPL11-eGFP (state A) and immobile RPL11-eGFP within induced inclusions (state B) are indicated. (D) Similar experiment as in A, but cells were treated with MG132 for the indicated time periods. # indicates RPL11-eGFP localizing within the nucleolar area (state A), while * indicates RPL11-eGFP within the stress-induced inclusions (state B). The experiments in (A, D) were performed at the same time and the same panels for the control conditions are presented. (E) Fluorescence recovery kinetics of the bleached regions in (D) were quantified and presented as the median relative intensity of 15 different cells. Scale bar 5 µm. Source data are available online for this figure.

transport, ribosome biogenesis, DNA replication, and the UPS system are enriched within the stress-induced inclusions (Fig. 3I). We identified 49 RPs, ubiquitin, and the HUWE1 E3-ligase, consistent with the imaging analysis of previous (Maghames et al, 2018) and presented studies, showing that these proteins accumulate in MG132-induced nuclear inclusions (Fig. 3I). Having established the proteome of MG132-induced inclusions, we performed a similar experiment to monitor the inclusion proteome composition during the recovery period. We mixed MG132-treated cells (heavy) with cells after 8 h of recovery (light) 1:1 before inclusion isolation and mass-spectrometry quantitation as before. We found that the abundance of 2425 aggregated proteins is reduced upon 8 h of recovery, including 43 RPs, ubiquitin, and HUWE1 (Fig. 3I). The proteomic analysis indicates that proteins within the MG132-induced inclusions are collectively eliminated during the recovery period at proteome-wide level, consistent with the imaging analysis monitoring individual RPs as model substrates. To further validate our findings and to test the specificity of the elimination process for the induced nuclear inclusions, we monitored changes in RPs levels in the cytoplasm and in the inclusions during the stress response. For this, we followed the fractionation protocol used for the isolation of nuclear inclusions that also provide cytoplasmic and nuclear fractions. (Methods). We chose RPs as model substrates as they are found both in the nucleolus-related inclusions and in the cytoplasm as part of the ribosome. Consistent with the imaging and proteomic analysis, MG132 increases RPs levels in the isolated inclusions, which is reduced almost to background control levels during the recovery period (Fig. 3J,K). In contrast, no changes in RPs levels were observed in the cytoplasm under these conditions (Fig. 3J). The combination of imaging, proteomics and biochemical fractionation strongly supports the concept of an elimination process during the recovery period for nuclear inclusions.

## Active nuclear proteasomes are recruited within nucleolus-related inclusions and promote their elimination

Previous studies proposed that nuclear inclusions can be exported and processed in the cytoplasm by the autophagy/lysosomal-vacuole pathways (Sontag et al, 2023; Iwata et al, 2005; Rideout et al, 2004). Despite the fact that the photoconversion analysis strongly indicates that inclusions are eliminated within the nuclear compartment, we experimentally tested the role of the autophagy pathway in this process. Using several approaches to block autophagy-mediated degradation, including the use of the lysosomal acidification inhibitor Bafilomycin A1 or the use of MEFs with a deletion of the autophagy-related E1-activating enzyme ATG7, we found that inclusion

elimination during recovery is independent of the autophagy system (Komatsu et al, 2005) (Appendix Fig. S3A–D).

We next assessed the role of the proteasome in the elimination process. For this, we used a concentration of MG132 (0.5 µM) that has been established to reduce proteasome activity (Appendix Fig. S4A) in the absence of generalized stress induction (Meriin et al, 1998), including a global increase in the ubiquitin/NEDD8 conjugates or the generation of nuclear inclusions (Fig. 4A; Appendix Fig. S2A) (Meriin et al, 1998). The elimination of stress-induced inclusions was blocked in the presence of low doses of MG132 added during the recovery period (Fig. 4A,B), indicating the critical role of the proteasome in the elimination of nuclear inclusions. As proteasomal degradation can occur both in the cytoplasm and the nucleus we tested the possibility that misfolded proteins in nuclear inclusions can be exported and degraded by cytoplasmic proteasomes. For this, we inhibited the key nuclear export receptor CRM1, with Leptomycin B (LMB) (Nishi et al, 1994; Laín et al, 1999). As shown in Appendix Fig. S4B–E, the recovery process is independent of the CRM1-mediated nuclear export, suggesting that the nuclear proteasome is involved in the elimination of the stress-induced nucleolus-related inclusions.

Previous quantitative mass-spectrometry data identified proteasome subunits within HS-induced protein aggregates (Maghames et al, 2018). However, it is not clear whether this represents misfolded proteasomal subunits that accumulate within the generated nucleolus-related inclusions or active proteasomes that are recruited for the elimination of misfolded proteins.

To monitor the distribution of active proteasomes during the stress and recovery period we labeled cells with a proteasome activity-based probe (ABPs). ABPs are small molecules consisting of a proteasome inhibitor linked to a small fluorophore which specifically labels active proteasomes in fixed and live cells (Gan et al, 2019). In unstressed cells, ABP is uniformly distributed within the nucleus (Fig. 4C). Upon stress (HS or MG132) we observed the accumulation of the ABP signal within the formed inclusions, also labeled by ubiquitin (Fig. 4C; Appendix Fig. S4F). Interestingly, the ABP signal follows the same kinetics as the ubiquitin staining during the response: it is maintained within ubiquitin-labeled inclusions during the initial stages of recovery and disappears when inclusion elimination is completed (Fig. 4C; Appendix Fig. S4F). The use of high doses of the irreversible proteasome inhibitor lactacystin, shows that the localization of the ABP probe is not due to unspecific inclusion binding or lipophilicity, but rather due to specific labeling of active proteasomes (Fig. 4D). In addition, the presence of active proteasome upon 15 h MG132 treatment reveals that the used concentrations (5 µM), can induce proteotoxic stress and the generation of protein inclusions without fully inhibiting proteasome activity (Appendix Fig. S4F). The role of the proteasome in inclusion formation and recovery, was further

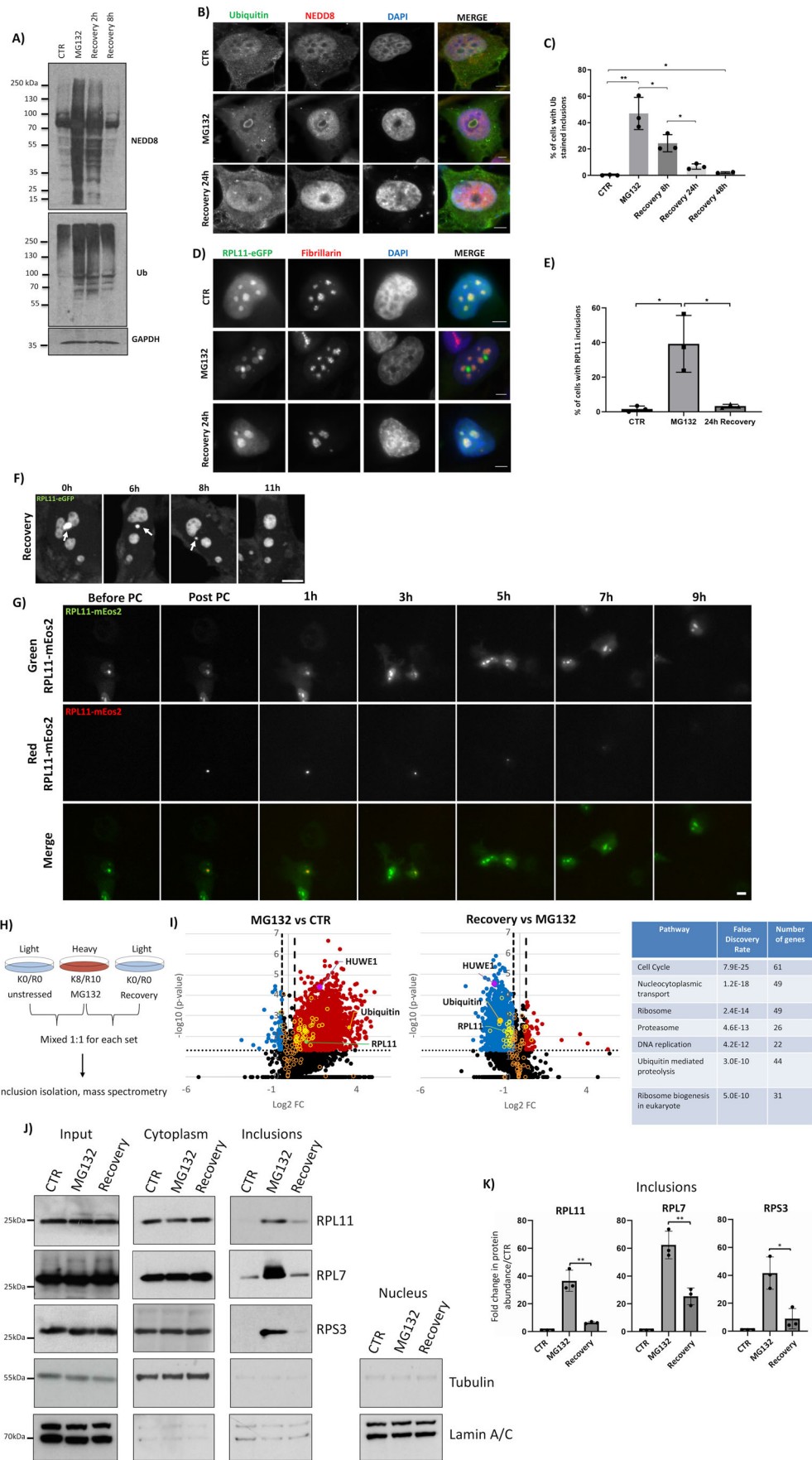

**Figure 3. Nucleolus-related inclusions are eliminated during the recovery period.**

(A) Extracts from H1299 cells treated with MG132 (5 µM, 15 h) before allowed to recover as indicated, were used for western blot analysis. GAPDH is used as a loading control. (B) H1299 cells treated as indicated were fixed and stained with ubiquitin (green) and NEDD8 (red). (C) Quantitation of the experiment in (B) showing the % of cells with ubiquitin-stained nuclear inclusions. *$P \leq 0.05$, **$P \leq 0.01$ by Student $t$-test. (D) Similar experiment as in (B), performed in H1299 RPL11-eGFP cells, stained with fibrillarin (red). Nuclei were stained with DAPI (blue). Scales bar 5 µm. (E) Quantitation of the experiment in (D) showing the % of cells with RPL11-eGFP within inclusions, based on its separation from fibrillarin. For all graphs, ~100 cells were counted per condition and values from three independent experiments are represented as mean with ±S.D. *$P \leq 0.05$ by Student $t$-test. (F) Live-imaging performed on H1299 RPL11-eGFP cells during MG132 treatment. White arrows indicate the position of nucleolus-related inclusions. Scale bar 10 µm. (G) Photoconversion experiment was performed as described in Methods. Green RPL11-mEos2 was photoconverted within MG132-induced inclusions and the fate of the red RPL11-mEos2 was followed during recovery for the indicated time period. Scale bar 10 µm. (H) Schematic representation of the SILAC approach for analysis of proteome changes in MG132-induced inclusions or during the recovery period. (I) Volcano plots of SILAC-based quantitative analysis. The left plot is the quantitation of MG132-treated vs control untreated cells, and the middle panel recovery of 8 h vs MG132-treated cells. The y-axis represents the −log10 of the $p$ value (Student $t$-test), whereas the x-axis is the log2 of the mean values of peptide abundance ratio from three independent experiments. In yellow are the identified RPs. Ubiquitin (protein group corresponding to the three ubiquitin precursors), RPL11, and HUWE1 are also indicated in the plots. Thresholds represented by dashed lines correspond to log2 FC and $p$ value of ±0.378 and 0.05, respectively. The right panel represents a group of proteins that are enriched in stress-induced inclusions. The top seven groups were selected, and the false-discovery rate is indicated. Gene Ontology analysis was performed using ShinyGO 0.80. (J) 3 × 10 cm dishes of 80% confluent H1299 cells were either unstressed (CTR), stressed with MG132 (5 µM, 15 h), or stressed and allowed to recover for 8 h. Cells were used for fractionation as described in Methods. Fractions were analysed by western blotting against the indicated proteins. The same time of ECL exposure was employed for all western blot analysis for the different fractions. (K) Quantitation of the experiment performed in (J) for the isolated inclusions. Band intensity was quantified by ImageJ. Graphs represent the mean ± S.D. of the fold change in protein abundance compared to CTR. Each dot represents an independent experiment ($n = 3$). *$P \leq 0.05$, **$P \leq 0.01$ by Student $t$-test. Source data are available online for this figure.

assessed using epoxomicin, which, in contrast to MG132, is a highly specific and irreversible inhibitor of the proteasome (Meng et al, 1999). We found that similarly to MG132, Epoxomicin induces the formation of nucleolus-related inclusions in a dose-dependent manner (Appendix Fig. S4G). However, as Epoxomicin is an irreversible inhibitor no recovery of formed inclusions was observed even up to 24 h post-stress recovery period (Appendix Fig. S4H). This further emphasizes the important role of proteasome activity in the elimination of stress-induced nuclear inclusions.

## The ubiquitin pathway is dispensable for the formation but essential for the elimination of nucleolus-related inclusions

While ubiquitin and Ubls such as SUMO and NEDD8 are found on nucleolus-related inclusions as central components of the PQC system, their precise role in inclusion formation and/or elimination is currently unclear. An intriguing feature of the Ubl response to stress is that the induced conjugation of NEDD8 on proteins does not depend on the bona fide NEDD8 E1, E2 enzymes (referred to as canonical NEDDylation) but rather on enzymes of the ubiquitin system (UBA1, referred to as atypical NEDDylation) (Meszka et al, 2022; Santonico, 2020; Leidecker et al, 2012; Hjerpe et al, 2012) (Fig. 5A,B; Appendix Fig. S5A). A key characteristic of the response is the formation of hybrid ubiquitin/NEDD8 chains on target proteins (Meszka et al, 2022). To assess the role of ubiquitin and of the different modes of NEDD8 conjugation in inclusion formation/elimination, we used either a specific inhibitor of the ubiquitin-activating enzyme (MLN7243, UBAi) that blocks ubiquitination but also atypical NEDDylation or MLN4924 that specifically inhibits the NEDD8 E1-activating enzyme (NAE) and the canonical pathway (Fig. 5A,B). As shown in Fig. 5C,D, while UBAi does not affect inclusion formation, it blocks inclusion elimination when added during the recovery period. In contrast, the inhibition of NAE with MLN4924 (NAEi) (Fig. 5C,D; Appendix Fig. S5A) does not affect the elimination process, consistent with the idea that canonical NEDDylation is not involved in the nuclear stress response (Maghames et al, 2018).

To explore in more detail the role of the ubiquitin/Ubl system in the elimination of nucleolus-related inclusions, we targeted the

HUWE1 E3-ligase. HUWE1 was identified as the main E3-ligase that modifies RPs with ubiquitin but also with hybrid ubiquitin/NEDD8 chains upon stress, and it is recruited within the formed nucleolus-related inclusions (Fig. 3I) (Maghames et al, 2018; Sung et al, 2016). Knockdown of HUWE1 dramatically reduces elimination of nucleolus-related inclusions, at a similar level as inhibition produced by UBAi. This indicates that HUWE1 is the main E3-ligase of the ubiquitin/Ubl system that is required for inclusion elimination during the recovery period (Fig. 5E,F).

Molecular chaperones are important components of the PQC system as they can directly extract proteins from inclusions and/or participate in the refolding of proteins. HSP70 belongs to the HSP110/HSP70 disaggregation system and plays a key role in protein folding and in the disassembly of protein aggregates (Kohler and Andréasson, 2020; Reinle et al, 2022). We found that chemical inhibition of HSP70 by the specific inhibitor VER-155008 during the recovery period blocked the elimination of RPL11 inclusions (Appendix Fig. S5B,C). Collectively, the data indicate the cooperative action of HUWE1 E3-ligase with the HSP70 chaperone system to promote nuclear inclusion elimination through the proteasome.

## Pol I activity maintains RPL11 mobility and promotes the elimination of nucleolus-related inclusions

Our understanding of the PQC system has been revolutionized by the discoveries of noncoding RNAs (ncRNAs) as scaffold entities that regulate the dynamics of assembly and dissolution of protein inclusions (Latonen, 2019; Wang et al, 2019; Luo et al, 2021; Onoguchi-Mizutani and Akimitsu, 2022; Audas et al, 2012; Balcerak et al, 2019; Gao et al, 2022; Kim et al, 2023). Nucleolar activity and plasticity rely on the production of ribosomal RNA (rRNA) and other ncRNAs (Latonen, 2019; Feric et al, 2016; Hayes and Weeks, 2016; Dash et al, 2023). We assessed the role of Pol I in the dynamics for the formation and elimination of nucleolus-related inclusions. For this, we used low doses of Actinomycin D (5 nM, ActD) that preferentially block Pol I activity and rRNA production (Bailly et al, 2016; Perry and Kelley, 1970). A time course experiment indicated that 1 h treatment with ActD has no significant effect on nucleolar morphology, whereas, as expected,

**A)**

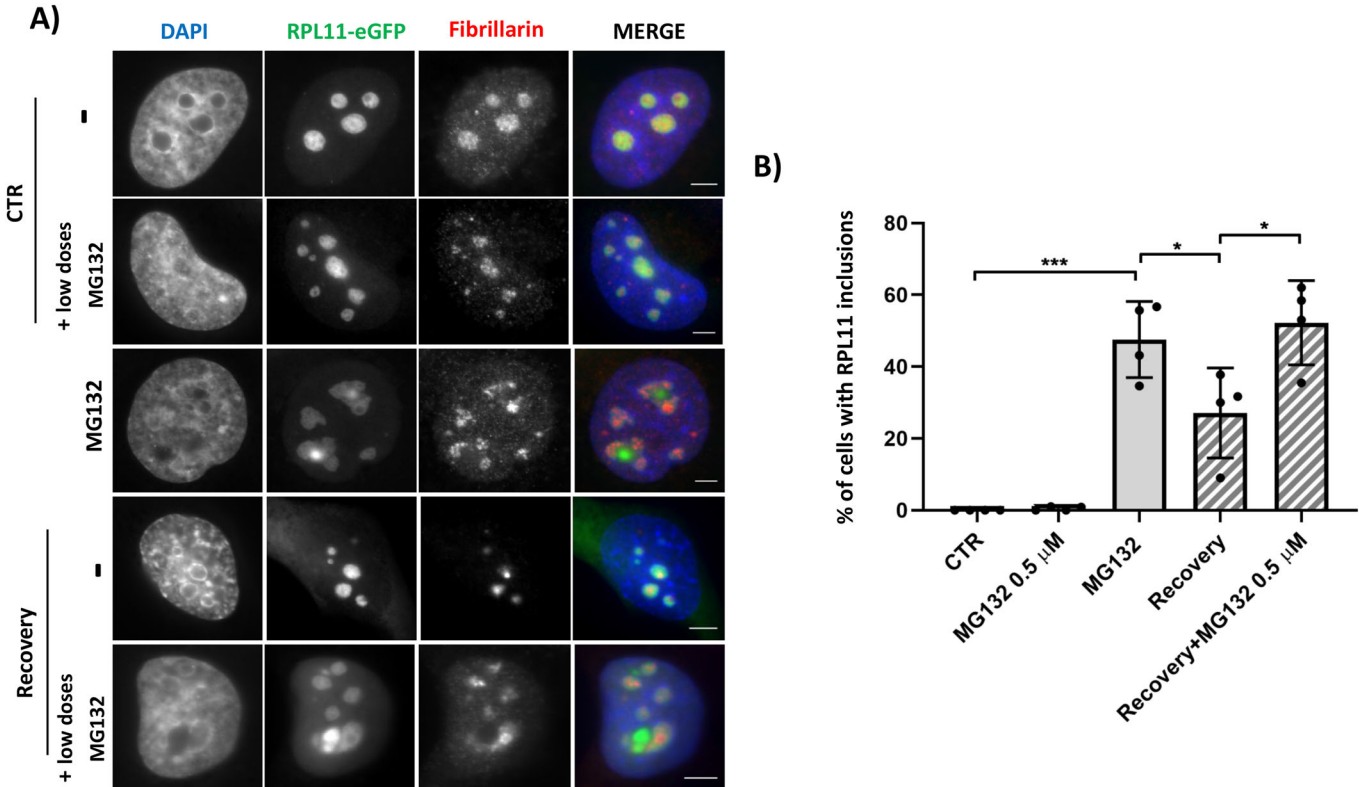

**B)**

**C)**

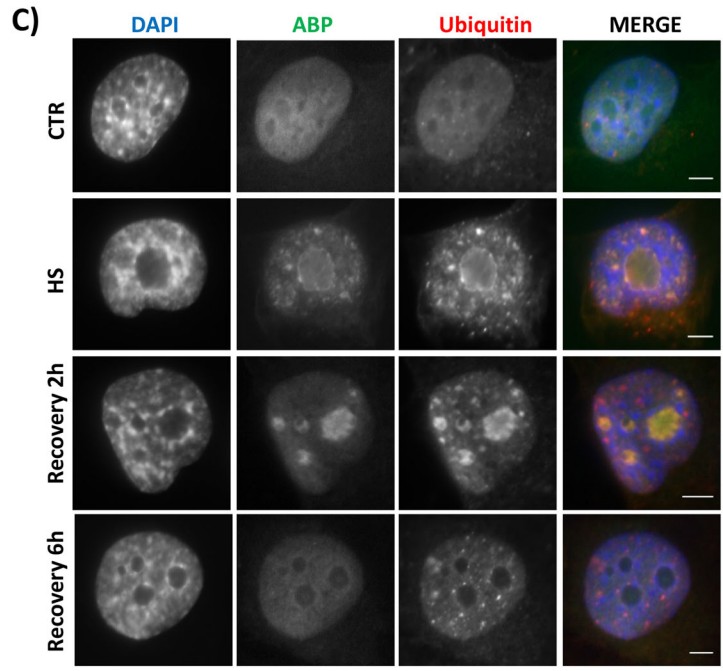

**D)**

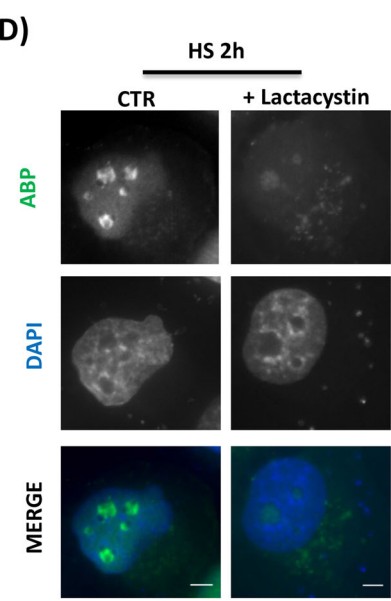

Figure 4.   Proteasome activity is required for the elimination of nucleolus-related inclusions during recovery.

(A) H1299 RPL11-eGFP cells were treated with MG132 (5 µM, 15 h) before recovery in the absence or presence of low doses of MG132 (0.5 µM) and stained for fibrillarin. Nuclei are stained with DAPI. Scales bar 5 µm. (B) Quantitation of the experiment in (A) showing the % of cells with RPL11 inclusions. Approximately 100 cells were counted per condition. Values represent the mean ± S.D. *$P \le 0.05$, **$P \le 0.01$, and ***$P \le 0.001$ by Student $t$-test. (C) H1299 cells were heat-shocked (43 °C, 2 h) before recovery at 37 °C. After fixation, active proteasomes were labeled by incubating cells with bodipy-FL-Ahx3L3VS (ABPs) for 30 min at RT (500 nM), before staining for ubiquitin (red). (D) Similar experiment as in (C), but lactacystin was used prior to ABPs incubation in order to block proteasome activity to test the specificity of the ABPs staining. Scales bar 5 µm. Source data are available online for this figure.

prolonged 15 h treatment caused nucleolar fragmentation, also referred to as nucleolar segregation (Fig. 6A).

To determine the role of Pol I in nucleolus-related inclusion formation and elimination, we decided to use conditions of acute inhibition (1 h ActD) to avoid any interference with the response due to nucleolar fragmentation upon prolonged Pol I inhibition. We found that 1 h ActD treatment prior to MG132-induced stress (Fig. 6B) almost completely blocked the elimination of MG132-induced inclusions (Fig. 6C–E). Strikingly, inhibiting Pol I exclusively during the 8 h of recovery period had no effect on the elimination processes, indicating that Pol I activity is required during the inclusion formation upon stress to ensure their elimination upon recovery (Fig. 6C,E). We obtained similar results using instead to ActD, the highly specific Pol I inhibitors CX-5461 or BMH-21 (Appendix Fig. S6A,B) (Drygin et al, 2011; Peltonen et al, 2014, 2010). To assess the correlation between nucleolus-related inclusion elimination and cell viability, we analysed more than 250 cells in live imaging over 28 h of recovery. During this period, aggregate elimination is severely compromised in ActD-treated cells compared to untreated cells (Fig. 6F). In addition, an increase in cell death was observed in ActD-treated cells (Fig. 6G). The above data suggest that specific Pol I products synthesized during the stress response assist in the dissolution of nuclear inclusions during the recovery period. Additionally, the inability of cells to eliminate the nuclear inclusions is associated with cell death.

A biophysical property that is related to the dynamicity and the potential of protein inclusions to be eliminated is the mobility of their constituent proteins (Mateju et al, 2017; Streit et al, 2022). Based on the finding that Pol I activity is essential for the elimination of nucleolus-related inclusions, we monitored by FRAP the effect of acute Pol I inhibition on the mobile fraction of RPL11 during inclusion formation upon stress (Olson and Dundr, 2005). We found that acute Pol I inhibition for 1 h, followed by MG132-induced stress, accelerated the transition of RPL11 to the immobile state (Fig. 6H; Appendix Fig. S6C). The data suggest that Pol I prevents the transition of nucleolar proteins into an immobile state during stress. This directly impacts the recovery process as it facilitates the elimination of inclusions.

## The Intergenic Spacer 42 Pol I-dependent lncRNA localizes within stress-induced inclusions and is required for their elimination

In addition to the production of rRNA as the key component of ribosome biogenesis, Pol I induces the expression of a series of lncRNAs localized within the intergenic spacer (IGS) region of rDNA (Pirogov et al, 2019; Jacob et al, 2013). The IGS lncRNAs are a unique class of lncRNAs (~300 nucleotides), as their sequence differs considerably from the rRNA sequence. Importantly, these

lncRNAs are expressed at very low levels in homeostatic conditions, but several of them (e.g., $IGS_{16}$/$IGS_{22}$/$IGS_{28}$) are induced upon HS and acidosis and participate in inclusion formation (Vydzhak et al, 2020; Jacob et al, 2013; Wang et al, 2018; Audas et al, 2012). Whether IGS lncRNAs control inclusion elimination and whether the HS/acidosis-induced IGS lncRNA also participates in MG132-induced aggregate dynamics is unclear. To better understand how Pol I activity controls MG132-induced inclusion elimination, we performed a systematic analysis of the expression of IGS lncRNAs within the rDNA IGS locus. Based on the finding that Pol I activity is required during stress for inclusion elimination during recovery (Fig. 6), we employed the following selection criteria to identify potential candidates that are required for the elimination process: (1) Expression of IGS lncRNAs should be induced upon stress (MG132) and (2) The induced expression should depend on Pol I activity. The analysis shows (Fig. 7) that in contrast to HS which was previously shown to induce the expression of several IGS lncRNAs(Audas et al, 2012, 2016), MG132 treatment has a more modest effect. Amongst tested lncRNAs, $IGS_{42}$ fulfilled the above set criteria and was selected for further analysis. To test the role of $IGS_{42}$ on inclusion elimination we transfected H1299 cells with siRNAs targeting $IGS_{42}$ (Appendix Fig. S7) and treated with MG132 before monitoring inclusion elimination during recovery by ubiquitin staining. Knockdown of $IGS_{42}$ reduces the % of cells that eliminate inclusions during the recovery period (Fig. 8A,B). This indicates that Pol I-dependent expression of IGS lncRNAs$_{42}$ during MG132-induced stress is at least part of the mechanism by which Pol I promotes efficient inclusion elimination during the recovery period. To assess the specificity of $IGS_{42}$ in the MG132-induced inclusion response, we tested the effect of knockdown of additional IGS lncRNAs in inclusion formation/elimination upon MG132 treatment. We chose $IGS_{16}$, $IGS_{18}$, and $IGS_{20}$ as this locus was previously proposed to control inclusion formation in response to other stress conditions, such as HS or acidosis (Wang et al, 2019; Audas et al, 2012). None of the tested IGS lncRNAs had a significant effect on the elimination of MG132-induced inclusions (Appendix Fig. S7). Additionally, we tested the effect of knockdown of $IGS_{24}$, $IGS_{27.5}$, and $IGS_{28}$ in inclusion formation/elimination. These lncRNAs displayed a similar $IGS_{42}$ response qualitatively, i.e., induced upon MG132 treatment and their expression is Pol I dependent, but did not pass the significance $t$-test analysis. We found that transfection of siRNAs against $IGS_{24}$ and $IGS_{27.5}$ caused significant cell death, which was dramatically increased upon MG132 treatment that technically compromised any meaningful subsequent analysis. Similarly to what we observed for $IGS_{16}$, $IGS_{18}$, and $IGS_{20}$, the knockdown of $IGS_{28}$ did not significantly affect either the formation or the elimination of MG132-induced nuclear inclusions (Appendix Fig. S7). The data indicate that the role of these lncRNAs in inclusion formation/elimination may depend on the applied stress.

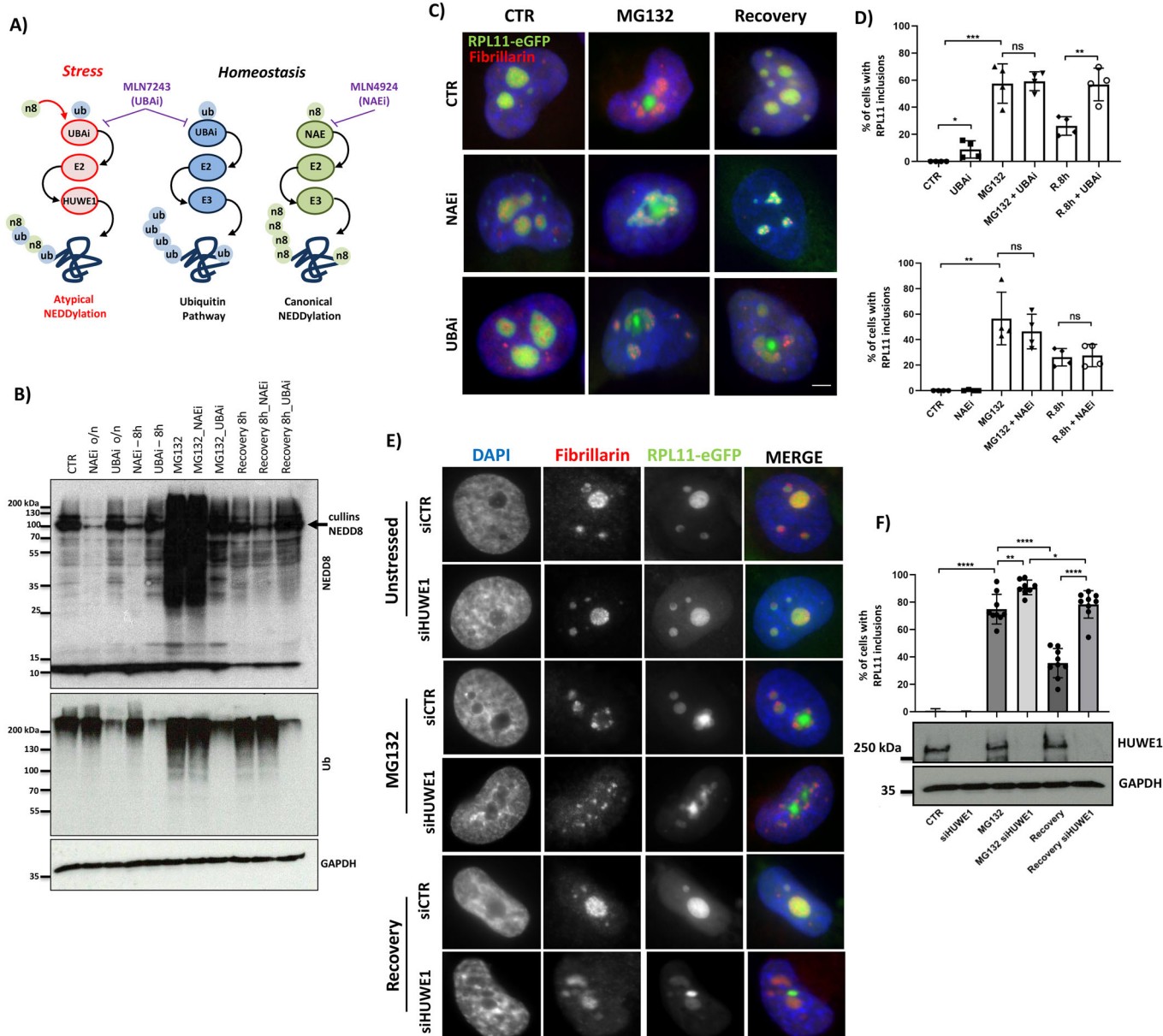

**Figure 5. The ubiquitin pathway and the HUWE1 E3 ligase are dispensable for the formation but essential for the elimination of nucleolus-related inclusions.**

(A) Schematic representation of the targets of used compounds. MLN7243 (UBAi) blocks the ubiquitin E1 enzyme UBA1, inhibiting both ubiquitination and the so-called atypical NEDDylation. MLN4924 (NAEi) blocks specifically the NEDD8 E1 enzyme (NAE) and the so-called canonical NEDDylation. (B) Western blotting showing the effect of UBAi inhibition (0.2 μM) on ubiquitination and atypical NEDDylation and NAEi (0.2 μM) on canonical NEDDylation (reduction in cullin NEDDylation-arrow). (C) Immunofluorescence merge image of H1299 RPL11-eGFP cells treated with UBAi or NAEi during stress (MG132, 5 μM, 15 h) or during the recovery period. (D) Quantitation of the experiment in (C). Approximately 100 cells per condition were used. Values represent the mean ± S.D. *$P \leq 0.05$, **$P \leq 0.01$, and ***$P \leq 0.001$ by Student $t$-test. (E) H1299 RPL11-eGFP cells were transfected with HUWE1 siRNA 48 h prior to be stressed with MG132 (5 μM, 15 h) and then recovered. Cells were stained with the nucleolar marker fibrillarin. (F) Quantitation of the experiment in (E). Values represent the mean ± SD in the corresponding graph. Each dot represents an independent experiment. Scale bar 5 μm. *$P \leq 0.05$, **$P \leq 0.01$, and ****$P \leq 0.0001$ by Student $t$-test. The bottom panel is a western blot analysis of one of the experiments in (E) with the indicated antibodies. Source data are available online for this figure.

To provide insights on the mechanism by which $IGS_{42}$ is involved in inclusion elimination, we monitored the localization of $IGS_{42}$ in H1299 RPL11-eGFP cells by FISH analysis using a fluorescent RNA probe against the $IGS_{42}$ lncRNA sequence. As expected, in unstressed cells, the levels for $IGS_{42}$ are relatively low

with diffused staining within the nucleolus. However, upon MG132 treatment while $IGS_{42}$ lncRNA is still detected within the nucleolus, a strong localization of $IGS_{42}$ within the stress-induced RPL11 inclusions was observed (Fig. 9A). This was further confirmed by applying two different high-resolution approaches, namely the 3D

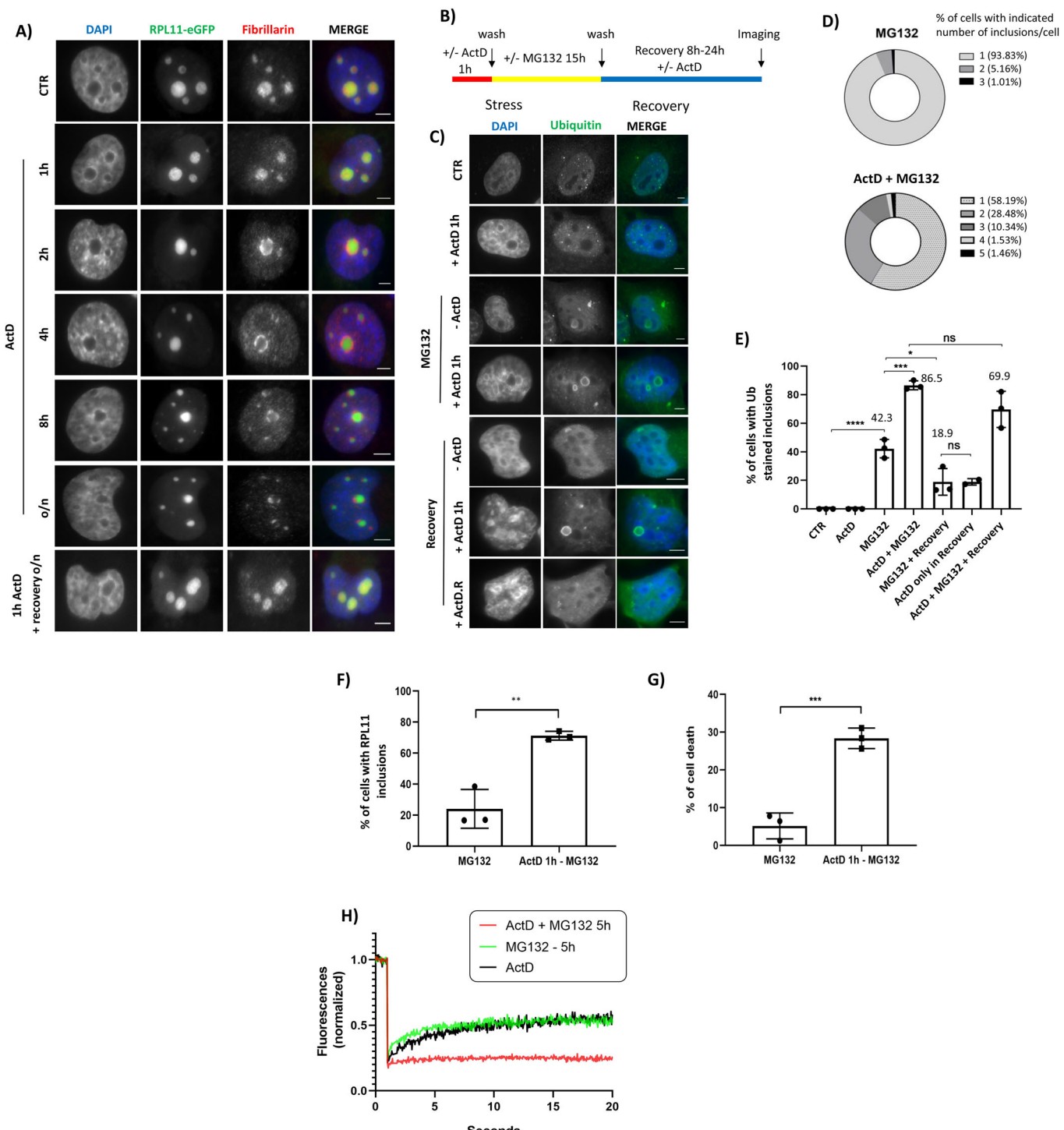

**Figure 6. RNA Polymerase I activity maintains the solubility of nucleolus-related inclusions and is required for inclusion elimination.**

(**A**) H1299 RPL11-eGFP cells were exposed for 1, 2, 4, 8, or 15 h (o/n) with Actinomycin D (ActD, 5 nM) before staining for fibrillarin (red). Scale bars 10 µm. (**B**) Schematic representation of the treatment protocol followed. (**C**) H1299 cells were treated with ActD (5 nM) for 1 h before exposed to proteotoxic stress (MG132, 5 µM, 15 h), or treated with ActD only during the recovery period (+ActD.R) and stained for ubiquitin (green). Scale bars 10 µm. (**D**) Quantitation of % cells with the indicated number of ubiquitin-stained inclusions in (**C**). (**E**) Quantitation of the % of cells with ubiquitin-stained inclusions under the indicated stress and recovery conditions. Values represent the mean ± SD. *$P \leq 0.05$, ***$P \leq 0.001$, and ****$P \leq 0.0001$ by Student $t$-test. (**F, G**) Cells were followed by live imaging up to 28 h post-stress, and the % of cells with RPL11 inclusions is presented (**F**) and the % of cell death (**G**). The experiment was performed three times, and the fate of around 100 cells per experiment was determined. Values represent the mean ± S.D. **$P \leq 0.01$, ***$P \leq 0.001$ by Student $t$-test. (**H**) Fluorescence recovery kinetics for RPL11 upon MG132 treatment for 5 h or when cells were treated with ActD for 1 h prior to MG132 treatment. Source data are available online for this figure.

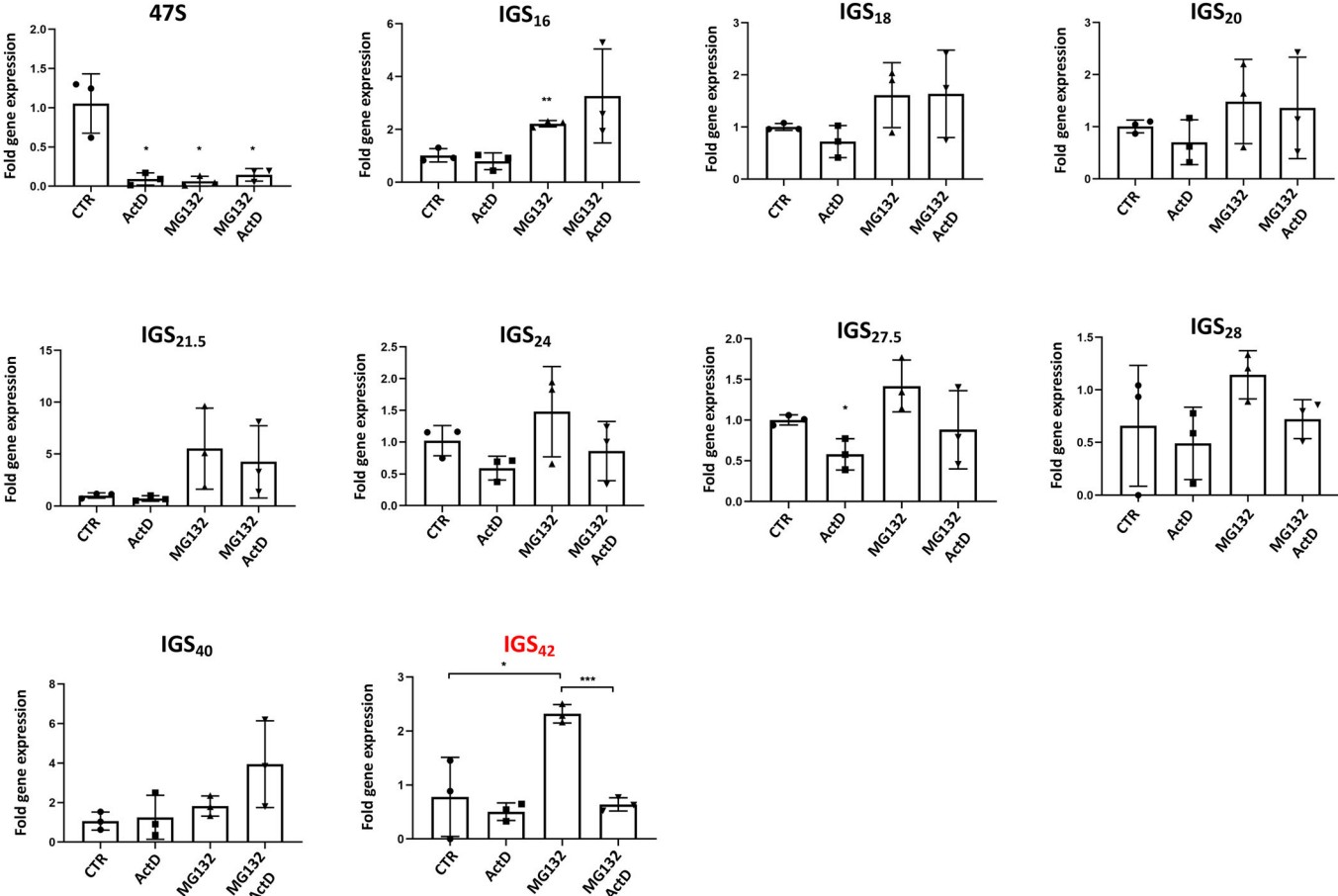

**Figure 7. Expression of IGS lncRNAs upon MG132-induced stress.**

H1299 cells were either untreated or treated with ActD (5 nM) for 1 h before washing and treated as indicated with MG132 (5 μM, 15 h). Extracts were used for RNA isolation and monitoring expression of the indicated RNAs by qRT-PCR as described in Methods. Values represent the average fold change in gene expression ±SD. For each independent experiment ($n = 3$), a triplicate was performed. *$P \leq 0.05$, ***$P \leq 0.001$ by Student *t*-test. In red, IGS$_{42}$ was selected for further analysis. Source data are available online for this figure.

reconstruction of images derived from confocal microscopy (Fig. 9B; Appendix Fig. S8A) and 3D random illumination microscopy (3D-RIM), which with intrinsic robustness to aberrations it allows accessing numerically, super-resolved reconstructions with high spatial resolution from low-resolution images by using random speckled illuminations (Appendix Fig. S8B) (Mangeat et al, 2021). Quantification of the mean signal intensity, shows that upon stress IGS$_{42}$ lncRNA is significantly enriched within the induced aggregates relatively to the nucleus/nucleolus (Fig. 9B). We also monitored the localization of pre-rRNAs derived from the processing of the 47S precursor rRNA, the main product of Pol I. We used RNA probes against the Internal Transcribed Spacers (ITS) located between 18S and 5.8S rRNA (ITS1) and between 5.8S and 28S (ITS2) to detect the precursors for the small and the large ribosomal subunits, respectively, and probes against the mature 18S rRNA. In striking contrast to the localization of IGS$_{42}$, ITS1, ITS2, and 18S were excluded from the RPL11 inclusions induced upon MG132 treatment (Fig. 9C,D; Appendix Fig. S8C). The data indicate distinct roles for different groups of Pol I RNA products and suggest that the induction of Pol I-dependent lncRNAs such as

IGS$_{42}$ upon stress is directly implicated in the dynamics of nucleolus-related inclusions.

## Discussion

Defining mechanisms that promote the elimination of protein inclusions is a key step toward understanding how cells deal with stress insults and maintain proteostasis. Here, we identify components of a PQC system that specifically operates in the nucleus for the elimination of nucleolus-related inclusions. The dramatic reorganization of the nucleolus upon stress is accompanied by the generation of inclusions that are specifically labeled with ubiquitin and Ubls, such as NEDD8 and SUMO, most likely as an indication of protein misfolding, resembling previously reported nucleolar aggresomes (Latonen, 2019; Latonen et al, 2011). Our studies now reveal that the critical role of the ubiquitin/Ubl system is not during the formation of these inclusions but rather during the recovery period for the elimination process. This notion is strengthened by the identification of HUWE1 as the main E3-ligase involved in inclusion elimination, defining a specificity factor of the ubiquitin/Ubl system in

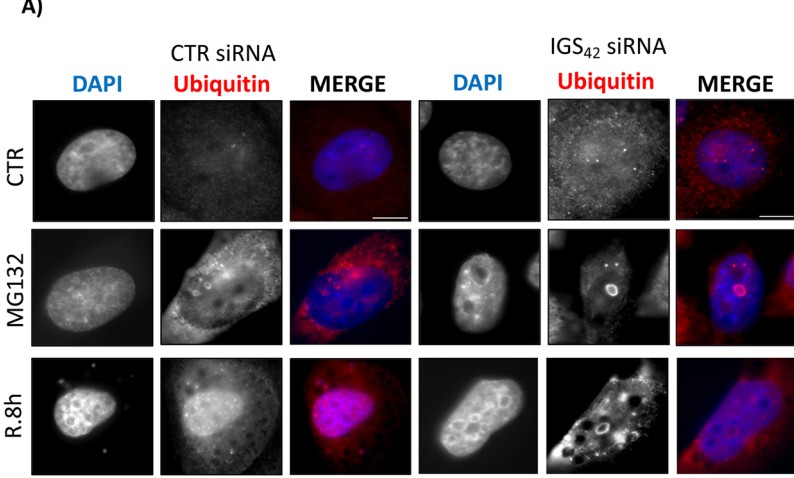

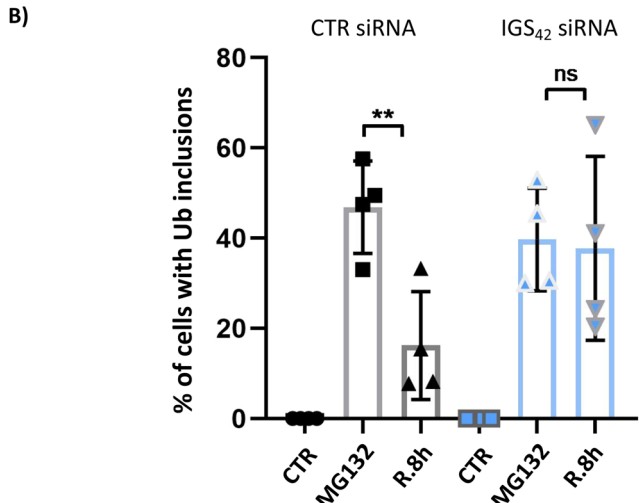

**Figure 8. IGS$_{42}$ lncRNA is required for nucleolus-related inclusion elimination during recovery.**

(A) H1299 cells were transfected with either control or siRNAs targeting IGS$_{42}$ as indicated. 48 h post-transfection cells were either untreated or treated with MG132 (5 μM, 15 h) before recovery of 8 h (R.8 h). Inclusion formation was monitored by ubiquitin staining. Scale bars 10 μm. (B) The graph represents the % of cells with ubiquitin-stained nuclear inclusions ±SD. In each independent experiment ($n = 4$) inclusions in ~100 cells were counted. **$P \leq 0.01$ by Student $t$-test. ns non-significant. Source data are available online for this figure.

the recovery process. Intriguingly, HUWE1 knockdown does not affect global ubiquitination or the ubiquitin decoration of the induced inclusions, most likely due to the redundant action of other E3-ligases involved in the stress response (Maghames et al, 2018). On the other hand, HUWE1 is essential for the modification of misfolded proteins with hybrid chains composed of ubiquitin/NEDD8 and possibly SUMO-2 (Maghames et al, 2018; Lobato-Gil et al, 2020). This raises the possibility that Ubls such as NEDD8 and/or SUMO-2 generate hybrid chains with ubiquitin during the stress response that operate as specific signals of the nuclear PQC system for the elimination of inclusions during recovery. Alternatively, as RPs are the major substrates of HUWE1 during the stress response, this may represent a specific role

for HUWE1 in the elimination of RPs-containing inclusions (Maghames et al, 2018; Xu et al, 2016; Sung et al, 2016; Martínez-Férriz et al, 2022). Additionally, HUWE1 may not be exclusively operating in the nucleus for aggregate elimination, as recent data indicate that HUWE1 may also promote the elimination of cytoplasmic stress-induced aggresomes (preprint: Zhou et al, 2023). Collectively, the studies reveal a critical and potentially broad role for HUWE1 in protein aggregation management as a component both of the cytoplasmic and nuclear PQC systems.

Based on the differential mobility of RPL11 depending on its localization, either within a nucleolus-like compartment or within the formed inclusions (Fig. 2E), it is possible that during stress, damaged RPs and other nucleolar components transit from a liquid-liquid to a

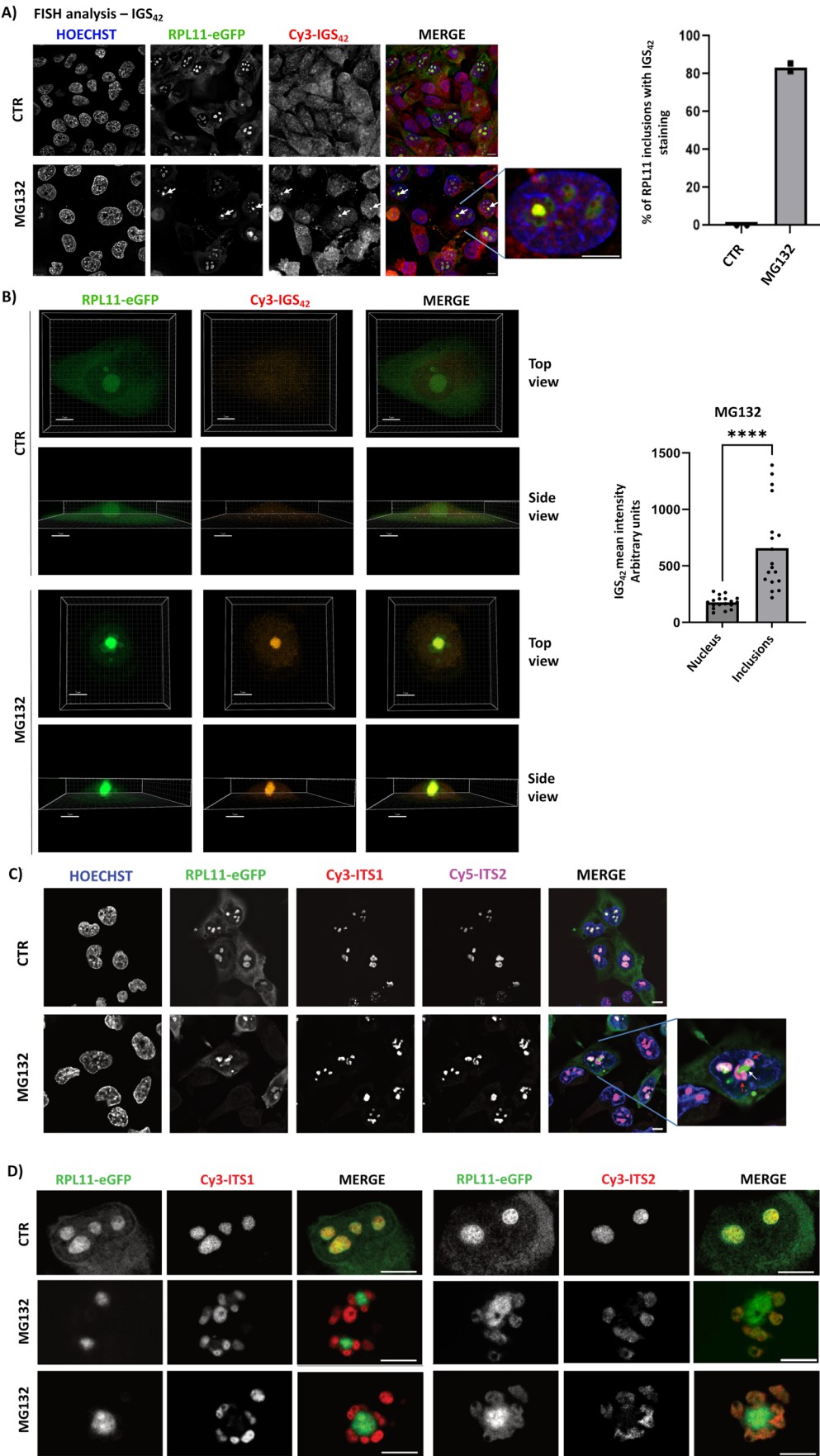

**Figure 9. IGS$_{42}$lncRNA is localised within MG132-induced nucleolus-related inclusions.**

(A) H1299 cells stably expressing RPL11-eGFP were either untreated or treated with MG132 (5 µM, 15 h). Cells were used for FISH analysis with the indicated probes as described in Methods. Right panel. Quantitation of the experiment performed in (A). Values represent the % of RPL11 inclusions co-stained with the IGS$_{42}$ probe ($n = 2$, ~100 cells/experiment). White arrows indicate stress-induced RPL11 inclusions (B). 3D reconstructions of confocal images from a similar experiment as in (A), for the top and side view. Scale bar 7 µm. The right graph represents the mean of the IGS$_{42}$ signal intensity in MG132-treated cells, calculated for the whole nucleus or within the induced inclusions as described in Methods. Each dot is an individual cell. ****$P \leq 0.0001$ by Student $t$-test. (C) A similar experiment as in (A), using instead probes against ITS1 and ITS2. White and red arrows indicate stress-induced RPL11 inclusions and ITS1/2 staining, respectively. (D) High-resolution images (3D-RIM) of the experiment performed in (C). Scale bars 10 µm. Source data are available online for this figure.

liquid-solid state. This may represent a protective mechanism to preserve nucleolar function during stress and/or during the recovery period. Consistent with this hypothesis, the photoconversion experiment strongly indicates that in the majority of cells, RPs within inclusions are eliminated during recovery and do not refold back into an active nucleolus. Previous studies using relatively mild stress conditions indicated that misfolded proteins within inclusions can escape degradation and retain their functionality during recovery (Wallace et al, 2015; Saad et al, 2017; Nollen et al, 2001; Ali et al, 2023). The extent of inclusion elimination and the fate of misfolded proteins (refolding vs elimination) may thus depend on the type/severity of the applied stress. Severe or prolonged stress conditions can cause the switch of a refolding mechanism into degradation, as the preferred fate of resident proteins within inclusions.

The dynamic recruitment of active proteasomes from the nucleoplasm within the induced inclusions kinetically coincides with their ubiquitin/Ubl labeling, defining the ubiquitin/Ubl pathways and nuclear proteasomes as the key machineries that cooperatively execute inclusion elimination. The process also strictly depends on the activity of the HSP70 chaperone. As an established component of the PQC system, HSP70 may participate in the extraction/disaggregation of misfolded proteins from the inclusions, mediating their shuttling and subsequent processing by the proteasome (Wang et al, 2018, 2019; Gallardo et al, 2020; Nollen et al, 2001; Parsell et al, 1994; Goloubinoff et al, 1999; Reeg et al, 2016). Additionally, HSP70 may also assist in the recruitment of the proteasome within the induced inclusions, as shown for the HSPA1 chaperone in 26S proteasome recruitment at active translating ribosomes upon HS (Tian et al, 2021).

While we cannot exclude the existence of a pathway that mediates the export and processing of nuclear inclusions in the cytoplasm (Rose and Schlieker, 2012), the data indicate that such a process should be autophagy/lysosome and CRM1 export receptor-independent. Additionally, the timing of inclusion elimination (~50% within 8 h) also indicates that the recovery process is independent of nuclear membrane breakdown during the cell cycle. Collectively, the data strongly support the concept of a proteasome-dependent process that operates within the nucleus for the elimination of inclusions.

The activity of RNA Pol I was also revealed as an additional critical element of the recovery process. The reduction in rRNA synthesis during stress, is believed to increase the susceptibility of RPs to misfolding and their transition into immobile inclusions (Lam et al, 2007; Warner, 1977). This process may be facilitated by the specific induction of Pol I or Pol II dependent IGS lncRNAs during stress, as previously demonstrated for IGS$_{16}$/IGS$_{22}$/IGS$_{28}$ or lncRNA PAPAS upon HS and acidosis (Wang et al, 2019; Audas et al, 2012; Santoro et al, 2010; Zhao et al, 2018; Feng et al, 2023). Our study now reveals that the induction of such lncRNAs (IGS$_{42}$) upon stress is critical during the recovery period and inclusion

elimination. Additionally, the expression and the role of IGS lncRNAs in inclusion dynamics may be stress-specific. Collectively, the data support a model where the IGS lncRNAs induced during stress, while facilitating the formation of nuclear inclusions, they also maintain the required biophysical properties of resident proteins (RPs mobility) to permit their elimination during recovery. This is supported by the observation that Pol I inhibition blocks inclusion elimination only if employed prior to stress but not during the recovery period, consistent with the idea that the expression of IGS lncRNAs during stress pre-defines the elimination process. Critically, the finding that the lncRNA IGS$_{42}$ is enriched within the generated inclusions upon stress, suggests that the role of such lncRNAs on inclusion dynamics is direct, potentially through the scaffolding function of RNAs during inclusion formation and elimination. While our studies have revealed that IGS$_{42}$ is required for efficient nuclear inclusion clearance, it is very likely that other RNA Pol I-dependent lncRNAs participate in the elimination process.

In coordination with the UPS-Chaperone systems, Pol I-dependent lncRNAs represent essential elements for the function of the nucleolus as a detention center; to buffer aggregation of misfolded proteins in the nucleus during the stress response ensuring their elimination during recovery (Lafontaine et al, 2021; Frottin et al, 2019; Audas et al, 2016). A detailed understanding of essential modules of the PQC system that operate in the nucleus may lead to the design of approaches for the elimination of nuclear inclusions observed in proteinopathies, which is anticipated to re-establish normal cellular function.

## Methods

### Reagents and tools table

| Reagent/resource | Reference or source | Identifier or catalog number |
|---|---|---|
| **Experimental models** | | |
| H1299 large-cell lung carcinoma | ATCC | CRL-5803 |
| H1299 RPL11-GFP | Sundqvist et al, 2009 | |
| ATG7 MEFs | Komatsu et al, 2005 | |
| **Recombinant DNA** | | |
| RPL7-eGFP | This study | Methods |
| RPS7-eGFP | This study | Methods |
| iRFP-Nucleolin | This study | Methods |
| RPL11-mEos2 | This study | Methods |

| Reagent/resource | Reference or source | Identifier or catalog number |
|---|---|---|
| **Antibodies** | | |
| Rabbit anti-ubiquitin (Western) | DAKO | Z0458 |
| Mouse anti-ubiquitin (FK2) (immunostainings) | Viva biosciences | VB2500 |
| Mouse anti-fibrillarin | Abcam | ab4566 |
| Mouse anti-GAPDH (6C5) | Abcam | ab8245 |
| Mouse anti-GFP | Abcam | ab1218 |
| Rabbit anti-RPL7 | Abcam | ab72550 |
| Rabbit anti-RPS3 | Abcam | ab128995 |
| Rabbit anti-NEDD8 Y297 | Abcam | ab81264 |
| Mouse anti-tubulin | Cell Signaling | mAb#3873 |
| Rabbit polyclonal anti-HUWE1 | Bethyl laboratories | A300-486A |
| Mouse anti-Pol I (RPA194) | Santa Cruz | sc-48385 |
| Mouse anti-Lamin A/C (636) | Santa Cruz | sc-7292 |
| Goat anti-RPL11 | Santa Cruz | sc-25931 |
| AF488 anti-mouse | Life Technologies | A10680 |
| AF546 anti-rabbit | Life Technologies | A11003 |
| HRP anti-mouse | Sigma-Aldrich | A4416 |
| HRP anti-rabbit | Sigma-Aldrich | A0545 |
| HRP anti-sheep (used for goat antibodies) | Sigma-Aldrich | A3415 |
| **Oligonucleotides and other sequence-based reagents** | | |
| Primers | This study | Table EV1 |
| siRNAs | This study | Table EV1 |
| FISH probes | This study | Table EV1 |
| **Chemicals, enzymes, and other reagents** | | |
| MLN4924 | Active Biochem | A-1139 |
| MLN7243 | Chemieteck | CT-M7243 |
| Leptomycin B | Sigma-Aldrich | L2913 |
| Actinomycin D | Sigma-Aldrich | A4262 |
| Bafilomycin A1 | Sigma-Aldrich | B1793 |
| Lactacystin | Sigma-Aldrich | L6785 |
| VER-155008 | Sigma-Aldrich | SML0271 |
| BMH-21 | Sigma-Aldrich | SML1183 |
| Epoxomicin | Sigma-Aldrich | 324801 |
| G418 | Sigma-Aldrich | G8168 |
| MG132 | Viva Bioscience | VB2204 |
| CX-5461 | ApexBio Technology | A8337 |
| FuGENE® 6 | Promega | E2691 |
| Lipofectamine™ RNAiMAX | Thermo Fisher Scientific | 13778075 |

| Reagent/resource | Reference or source | Identifier or catalog number |
|---|---|---|
| **Software** | | |
| ALgoRIM | | ALgoRIM https://github.com/teamRIM/tutoRIM |
| FIJI | | https://imagej.net/software/fiji/ |
| easyCLEM | | https://www.nature.com/articles/nmeth.4170 |
| MaxQuant (version 2.1.0.0) | | https://www.maxquant.org |
| ImageJ | | https://imagej.net/ij/ |
| **Other** | | |

## Cell culture

Human large-cell lung carcinoma H1299 cells and ATG7 MEFs were cultured in Dulbecco's modified Eagle's medium supplemented with 10% fetal bovine serum (FBS) and standard antibiotics (Penicillin 50 U/ml and Streptomycin 50 μg/ml) in 5% $CO_2$ at 37 °C in a humidified incubator. H1299 cells stably expressing RPL11-eGFP (Sundqvist et al, 2009) were cultured in the presence of 500 μg/ml G418 for selection. Cells were cultured up to 90% confluency, then washed once with warm PBS, treated with Trypsin-EDTA (Promega), and split to the desired confluency. Cell lines were not authenticated but were routinely tested for mycoplasma contamination and kept in culture for a maximum of passage 15.

## Transfections

Cells were seeded in six-well plates to the desired confluency. About 5 nM of siRNAs (Table EV1) was transfected with Lipofectamine RNAiMAX according to the manufacturer's instructions. Non-target siRNA was used in control transfections. Experiment was performed 48 h post-transfection. RPL7 and RPS7 were cloned as N-terminal fusions to eGFP in the pEGFP-N1 plasmid (Clontech). The RPL11-mEos2 construct was generated by replacing the eGFP in the pEGFP-N1 RPL11-eGFP plasmid with mEos2. iRFP-Nucleolin plasmid was generated by cloning the iRFP cDNA into a pCMV-eGFPC1-nucleolin vector (Addgene, #28176) once the eGFP tag was removed by digestion with *KpnI* and *AgeI* restriction endonucleases. All constructs were confirmed by sequencing (Eurofins Genomics).

## Immunofluorescence microscopy

Twenty-four hours before HS/MG132 treatment, cells were seeded on round coverslips previously washed in EtOH. After the treatment, cells were washed once with PBS and then fixed with 4% formaldehyde (Sigma-Aldrich) for 10 min. Then, cells were washed three times with PBS and permeabilized with 1% Triton in PBS for 10 min. After another raw of washing with PBS, cells were blocked with a PBS solution containing 0.05% Tween + 1% Goat serum (Sigma-Aldrich) for 30 min and incubated with the primary antibody diluted in 0.05% Tween + 1% Goat Serum in PBS overnight at 4 °C. The day after, cells were washed three times in PBS containing 0.05% Tween and incubated with fluorescein

(FITC)-conjugated secondary antibody (1/500) diluted in 0.05% Tween + 1% Goat serum in PBS for 1 h at room temperature (RT) in the dark. Samples were washed 3x10min with PBS (0.05% Tween + 1% Goat Serum) and then stained with DAPI (1/20 000) for 30 s at RT. Cells were quickly washed twice with PBS and mounted with ProLong™ Glass Antifade Mountant (Invitrogen). Samples were kept 24 h in the dark at RT before observation under the microscope Zeiss Axioimager Z1 or Zeiss Axiolmager Z2 at the x100 oil lens. The images were analysed by ImageJ software.

## Activity-based proteasome (ABPs)

Bodipy-FL-Ahx3L3VS (ABP) was kindly provided by Ovaa Huib's lab. Cells were seeded on round coverslips previously washed in EtOH. After the treatment, cells were washed once with PBS and then fixed with 4% formaldehyde (Sigma-Aldrich) for 10 min. Then, samples were washed three times with PBS and were immediately incubated at RT for 1 h with 500 nM Bodipy-FL-Ahx3L3VS diluted in PBS. Cells were washed once with PBS before being stained with DAPI (1/20,000) for 30 s at RT and mounted with ProLong™ Antifade Mountant (Invitrogen). Samples are kept 24 h in the dark at RT before observation under the microscope Zeiss Axioimager Z1 or Zeiss Axiolmager Z2. The images were analysed by ImageJ software. In order to ensure the high specificity of the Bodipy-FL-Ahx3L3VS probes, an irreversible proteasome inhibitor, Lactacystin (50 µM), was added before and during Bodipy-FL staining.

## Correlative-light electron microscopy (CLEM)

H1299 cells were grown on MatTek dishes with a finder grid. Cells were fixed with 0.05% glutaraldehyde (GA) and 4% paraformaldehyde (PFA) for 30 min at RT. Images were captured using an inverted Nikon TiE epi-fluorescence microscope equipped with a X100 objective lens (Apo, NA1.4, Oil), a SpectraX illumination system (Lumencore©) and a sCMOS camera (Hamamatsu© ORCA-Flash4.0), driven by NIS-elements. Cells were then fixed with 2.5% GA for 35 min in Sorensen buffer, pH 7.2, postfixed in 1% osmium in Sorensen buffer for 1 h, and treated with aqueous uranyl acetate 2% for 1 h at RT. The samples were then rinsed in water, dehydrated in an ethanol series, and embedded in Epon. Sections were cut on a Leica Ultracut microtome, and ultrathin sections were mounted on Formvar-coated slot copper grids. Finally, thin sections were stained with 2% uranyl acetate and lead citrate and examined with a transmission electron microscope (Jeol JEM-1400) at 80 kV. Images were acquired using a digital camera (Gatan Orius) at different magnifications. Alignments were performed using the easy-CLEM plugin managed by Icy Software: https://www.nature.com/articles/nmeth.4170.

## Expansion microscopy (ExpM)

The protocol was based on (Asano et al, 2018). H1299 cells or H1299 stably expressing RPL11-eGFP were seeded into round coverslips previously washed in EtOH. Cells were washed (PBS) and fixed with PFA 3.2% for 10 min at RT. After washing three times with PBS, cells were permeabilized with PBS-AT (PBS, Triton X-100 0.5%, and BSA 1%) for 1 h at RT. The primary antibody was incubated for 2 h at RT in PBS-BSA solution. The dilution of primary antibodies used here was five times higher compared to what we usually use for a "classical" immunostaining. For eGFP stable cell lines, anti-GFP was used to amplify the signal. Cells were washed three times with PBS-BSA prior to incubation with a secondary antibody for 2 h at RT. Once washed with PBS-BSA, the non-expanded samples were kept in PBS while the expanded samples underwent the expansion protocol. The stock solution AcX was diluted 1/100 in PBS immediately prior to addition to cells for 2 h at RT. Cells were then washed with PBS two times for 15 min at RT. A gelation chamber is performed as described in (Asano et al, 2018), and the stock solution was mixed with APS and TEMED and added dropwise onto cells for polymerization (1 h 30–37 °C). The gel was removed from the chamber and digested with proteinase K overnight at RT in the dark. The day after, the gel was expanded by washing it three times with water, and then observed under a confocal Leica SP5-SMD microscope.

## Live imaging

H1299 cells stably expressing RPL11-eGFP were seeded into 3.5 cm glass-bottom dishes (FluoroDish™, Ibidi, ref 81156) 24 h prior to transfection with iRFP-nucleolin plasmid. Cells at 90% confluency 48 h post-transfection were treated with MG132 (5 µM) for 15 h. Actinomycin D (5 nM) is added 1 h before MG132 as indicated to be washed prior to MG132 treatment. For the recovery experiment, cells were washed 3 times with warm PBS before adding fresh MG132-free DMEM into the well. Actinomycin D (5 nM) was added as indicated for 1 h, before cells were thoroughly washed with PBS (three times), fresh DMEM was added and stress was applied as indicated. Cells were visualized by confocal microscopy (Spinning Disk Dragonfly) either during the stress period or during the recovery period in a 37 °C and 5% $CO_2$ environmental chamber. In order to improve the detected fluorescent signal, we exchanged DMEM for a phenol red-free DMEM prior to imaging. A 60x plan apo lambda oil immersion lens with a 1.4 NA DT was used. Images were analysed by using ImarisViewer.

## Fluorescence recovery after photobleaching (FRAP)

Experiments were performed in H1299 cells stably expressing RPL11-eGFP. Cells were grown in glass-bottom 35 mm fluorodishes (Ibidi). After heat-shock (43 °C), MG132 (5 µM), and Actinomycin D (5 nM) treatment, cells were immediately mounted onto a ZEISS LSM780 multi-photonique confocal microscope equipped with an argon laser. Cells were kept in a chamber at 37 °C and 5% $CO_2$ during the acquisition. The 488 nm laser and a 63x Apo oil lens with 1.4 NA were used for imaging. A laser power of 0.7% was used in image acquisition, and the gain was adapted to the signal. The time for each image acquisition was 25 s which did not induce any bleaching with these acquisition parameters. To bleach the RPL11-eGFP protein, the RPL11-aggregate (characterized by high fluorescence intensity) or the RPL11 localized in the nucleolus compartment was scanned with maximum laser power (100%) for 100 iterations. The data were plotted in a graph to show recovery of fluorescence in the bleached area. For all data, the mean of 15 measurements is presented.

## Photoconversion

H1299 cells in 3.5 cm glass-bottom dishes (FluoroDish™) were transfected with a construct expressing RPL11-mEos2 (1 mg) with Fugene HD transfection reagent according to the manufacturer's instructions. 24 h post-transfection cells were treated with MG132

(5 mM, 15 h). The following day, cells were washed three times with PBS before adding fresh phenol red-free DMEM medium complemented with 10% FBS prior to imaging. The photoconversion experiments were performed with an Eclipse Ti Nikon microscope coupled to an iLAS2 RoperScientific module for laser manipulation in a chamber at 37 °C and 5% CO$_2$. A small area in the nucleolus was photoconverted with the 405 nm laser line, and cells were imaged with the 488 and 561 nm laser lines, with a 60×1.4 NA immersion oil objective, an EMCCD iXon 897 Ultra Andor camera (300 as EM Gain, 300 ms as exposure time). Images were acquired every 20 min.

## Fluorescence in situ hybridization

H1299-eGFP cells grown on glass coverslips were washed twice in PBS and fixed for 30 min with 4% paraformaldehyde in PBS (EMS). After washing twice in PBS, they were permeabilized for 18 h in 70% ethanol at 4 °C. The cells were then rehydrated twice for 5 min at RT in 2× SSC containing 10% formamide. Probe (Table EV1) hybridization was performed in the dark and at 37 °C for ≥4 h in a buffer containing 10% formamide, 2.1× SSC, 0.5 μg/ml tRNA, 10% dextran sulfate, 250 μg/ml BSA, 10 mM ribonucleoside vanadyl complexes, and 0.5 ng/μL of each probe. After two washes at 37 °C with 2× SSC containing 10% formamide, the cells were rinsed in PBS, counter-stained with Hoechst 33342 (Invitrogen), and mounted in Vectashield antifade mounting medium (Vector Laboratories). Observations were made with an inverted microscope (Ti-E/B; Nikon) equipped with an HG Intensilight illumination system and a CCD camera (Orca R2; Hamamatsu). The CFI Plan APO VC 100× oil objective (N.A. 1.4) was equipped with a motorized Piezo Nano Z100 TI, and stacks were acquired with a 0.2-μm-z step. Images were acquired with NIS Element software and deconvolution was performed with Huygens Deconvolution software (Scientific Volume Imaging). For super-resolution imaging, samples were analysed in confocal Zeiss LSM980 Airyscan 8Y, and 3D reconstruction was performed from z-stacks (31 stacks, acquired with a Nyquist step of 0.25 μm) using Imaris 10.1.1 (Bitplane an Oxford Instruments company).

3D random illumination microscopy (3D-RIM) was applied to H1299 cells stably expressing RPL11-eGFP relative to pre-rRNA labeling by FISH. The 3D-RIM homemade setup was coupled with an inverted microscope (TiE Nikon) equipped with a 100x magnification, 1.49 N.A. objective (CFI SR APO 100XH ON 1.49, Nikon) and two SCMOS camera (ORCA-Fusion, Hamamatsu) mounted in an industrial apochromatic alignment module (Abbelight SA). Fast diode lasers (Oxxius) with the wavelength centered at 488 nm (LBX-488-200-CSB) and 561 nm (LMX- 561L-200-COL) were used. The bandpass emission filters in front of the two respective cameras were FF01-514/30-25 for camera 1 and FF01-609/54-25 for camera 2. The binary phase modulator (QXGA fourth dimensions) conjugated to the image plane combined with polarization elements, was used to generate dynamic speckles on the object plane as previously described (Mangeat et al, 2021). The fluorescence was collected on one or two sCMOS cameras (OrcaFlash fusion) after passing through a Stop Line quad-Notch filter (Semrock NF03-405/488/561/635E-25), two relay lenses L4 and L5 (focal equal to 200 mm), and two bandpass filter (Semrock FF01-514/ 30-25 for GFP and Semrock FF01-609/54-25 for Cy3, respectively). The synchronization of the hardware (Z-platform, cameras, microscope, laser, and SLM) was driven by an upgraded version of the 22-commercial software INSCOPER. The acquisitions comprised 48 optimized random patterns for each plane. The image reconstructions were performed with the software (ALGoRIM https://github.com/teamRIM/tutoRIM) using the variance reconstruction method. The Wiener filter used was 0.9 (GFP channels), 0.8 (Cy3-IGS42 channel) or 0.9 (Cy3-ITS1 and Cy3-ITS2 channels), the deconvolution parameter was 0.9 (GFP channel), 0.8 (Cy3-IGS42 channel), or 0.5 (Cy3-ITS1 and Cy3-ITS2 channels) and the regularization parameter was 0.7 (GFP channel), 0.8 (Cy3-IGS42 channel), or 0.30 (Cy3-ITS1 and Cy3-ITS2 channels). Bleaching correction was done after RIM reconstruction with open-source FIJI software (https://imagej.net/software/fiji/) based on exponential FIT from the background signal. The mean signal intensity within the stress-induced inclusions and the nucleus was determined using ImageJ. The mean value represents the total intensity signal over the number of pixels within the defined area of interest.

## RNA isolation and qRT-PCR analysis

Total RNA was isolated from H1299 cells (6 cm dish) using the RNeasy kit (Qiagen), following the manufacturer's instructions. Briefly, RNA was isolated with QIazol Lysis reagent, followed by chloroform treatment and ethanol precipitation. RNA was quantified, and cDNA synthesis was carried out using QuantiTect Reverse Transcription kit RT-qPCR (Qiagen) was performed with 1 μg RNA. Quantitative PCR (qPCR) was performed using specific primers (Table EV1) for different IGS and SYBR Green master mix (Roche) on the LightCycler$^R$ 480 Real-Time PCR System and analyzed under LightCycler Software. The GAPDH gene was used as a reference housekeeping gene to calculate the DCt. Values represent the $2^{DDCt}$ for each tested IGS using the untreated/DMSO cells as a reference control condition. For each independent experiment, a triplicate was performed for each condition and the average Ct value was further used.

## Inclusion isolation and SILAC-based proteomics

For the isolation of cytoplasmic fractions, $7 × 10^6$ H1299 cells were resuspended in 300 μl of hypotonic buffer (20 mM Tris-HCl, pH 7.4, 10 mM KCl, 2 mM MgCl$_2$, 1 mM EGTA, 0.5 mM DTT, and 0.5 mM PMSF) and were incubated on ice for 5 min. Then, NP-40 was added to a final concentration of 0.1%. After 3 min incubation, samples were centrifuged at $1000 × g$ 4 °C for 5 min. The supernatant (cytoplasmic fraction) was mixed with an equal volume of 2x SDS Laemmli buffer (600 μl in total), and the pellet (nuclear fraction) was lysed in 600 μl of 2x SDS Laemmli buffer. Samples were then used for western blot analysis.

For the SILAC experiment, H1299 cells were labeled either with light (Lys0/ Arg0) or heavy (Lys8/Arg10) amino acids. An equal number of cells from each labeling condition ($3.5 × 10^6$ cells/condition) were mixed and snap-frozen before inclusion isolation. The protocol for the isolation of fractions enriched for nuclear inclusions was adapted from methods described for the isolation of *N*-lauroylsarcosine insoluble Alzheimer's disease aggregates and HS-induced inclusions (Diner et al, 2017; Maghames et al, 2018). In total, $7 × 10^6$ cells were resuspended in 1 ml of 20 mM HEPES, pH 8, 300 mM NaCl, 2 mM MgCl$_2$, Protease Inhibitors "c0mplete" (Sigma-Aldrich) and incubated on ice for 5 min before the addition of 0.1% NP-40 for 3 min as described above and centrifugation at 1000×g, 4 °C for 5 min. Nuclear enriched pellets were resuspended in 1 ml of 20 mM HEPES, pH 8, 300 mM NaCl, and 2 mM MgCl$_2$, Protease Inhibitors "c0mplete" and sonicated (Bioruptor pico, four cycles of 20 s). Membrane debris was eliminated by centrifugation at $200 × g$, 4 °C for 5 min. Supernatants were centrifuged at 15,000 × g, 4 °C

for 15 min. The pellet, enriched with nuclear inclusions and condensates, was resuspended in 1 ml of 20 mM HEPES, pH 8, 300 mM NaCl, 2 mM MgCl$_2$, 1% NP-40, protease inhibitors, 1% *N*-lauroylsarcosine before centrifugation at 200,000 × *g*, 4 °C for 30 min. The pellet containing insoluble inclusions was resuspended in 5% SDS, 50 mM triethylammonium bicarbonate, pH 8.5, and used for proteomic or western blot analysis.

## Mass-spectrometry analysis

About 10 µg of proteins from isolated *N*-lauroylsarcosine insoluble inclusions were digested in columns with Trypsin using the universal proteomics sample preparation kit ProtiFi according to manufacturer's instructions before eluted peptides were analysed by mass-spectrometry.

Tryptic peptides were resuspended in 10 µl of 0.2% formic acid and analyzed by nano-LC-MS/MS using nanoRS UHPLC system coupled to an Orbitrap Exploris 480 mass spectrometer using FAIMS Pro Duo interface (Thermo Fisher Scientific, Bremen, Germany). One microliter of each sample was loaded on an analytical C18 column (PEPMAP C18 2 µm 75 µm × 500 mm Thermo Fisher Electron) heated at 45 °C. The mobile phase flow rate was 300 nL/min, and 5% ACN + 0.2% FA in H$_2$Omq and 80% ACN + 0.2% FA in H$_2$Omq were used as buffers A and B, respectively. Peptides were eluted using a 2 h gradient: 0–25% buffer B for 102 min, 25–40% buffer B for 20 min, 40–90% buffer B for 2 min, and 90% buffer B for 5 min. The Orbitrap Exploris 480 was operated in FAIMS mode (gas flow of 3.9 L/min, two compensation voltages used: −45 and −60 v, 0.8 and 0.7 s cycle times per CV), and in data-dependent acquisition mode with the Xcalibur software. Survey scan MS spectra were acquired in the Orbitrap on the 375–1200 *m/z* range with the resolution set to a value of 60,000, AGC target was 300% with an IT of 50 ms. Following each survey scan, the most intense ions above a threshold ion count of 5e3 were selected for fragmentation by high-energy collision-induced dissociation at a normalized collision energy of 30%. The number of selected precursor ions for fragmentation was determined by the "Top Speed" acquisition algorithm. AGC target value was set at 100%, and automatic maximum injection time. The MS2 resolution was set to 15000. Isolation width was set at 1.6 m/z, and dynamic exclusion was used within 45 s to prevent repetitive selection of the same peptide.

Raw MS files were processed with MaxQuant software (version 2.1.0.0) for database search with the Andromeda search engine and quantitative analysis. Data were searched against Human entries in the UniProtKB protein database (UP000005640 download 2024_06) and the set of common contaminants provided by MaxQuant.

Carbamidomethylation of cysteines was included as a fixed modification, whereas variable modifications included, acetylation of protein amino (N)-termini, oxidation of methionine, glycine–glycine of lysine and heavy (Arg + 10, Lys + 8) isotope labeling. The specificity of trypsin digestion was set for cleavage after K or R, and two missed trypsin cleavage sites were allowed. The precursor mass tolerance was set to 20 ppm for the first search and 4.5 ppm for the main Andromeda database search. The mass tolerance in tandem MS mode was set to 20 mmu. The minimum peptide length was set to seven amino acids, and the minimum number of unique or razor peptides to one. Andromeda results were validated by the target decoy approach using a reverse database at both peptide and protein false-discovery rate of 1% at both the PSM and protein levels.

The "proteinGroups.txt" file produced by MaxQuant was further analyzed in Perseus (version 2.0.10.0). The SILAC ratios were log2-transformed and proteins from the reverse database and contaminants were removed. Statistical analysis was performed by employing a one-sample *t*-test (*p* value <0.05), when log2 SILAC ratios were compared against a value of 0 (control, log2(1)). Moreover, proteins with an average log2 SILAC ratio ≥0.378 or ≤−0.378 (corresponding to a ratio ≥1.3 or ≤0.77, respectively) were considered as high confidence variant proteins. The use of SILAC ratio cut-off plus the *p* value <0.05 as criteria instead of applying multiple testing correction could help reduce the false positive without excluding true-positive interacting partners in quantitative proteomics.

## Western blot analysis

For all samples, we routinely used ~20 µg of total protein. Proteins were resolved with NOVEX 1.0 mm Cassettes (life technologies) and by using 8 to 14% gels. Gels were transferred onto nitrocellulose membrane (Amersham™ Protran™ 0.2 µm NC) before blocking with 5% milk solution (PBS with 5% skimmed milk and 0.1% Tween-20, boiled and filtered) for 1 h at RT. Membranes were incubated with primary antibodies in TBS 0.1% Tween-20 with 3% BSA and 0.1% NaN$_3$ overnight at 4 °C. Membranes were washed 3 × 15 min with PBS 0.1% Tween-20 prior to incubation with the corresponding secondary antibodies diluted in 5% milk solution, at RT for 1 h. Membranes were washed 3 × 15 min with PBS 0.1% Tween-20 before incubation with ECL Western Blotting Detection Reagents (Amersham) and exposure to medical films (Konica Minolta). Band intensity was quantified by ImageJ.

## Data availability

The raw data regarding the SILAC-based proteomic analysis are submitted to ProteomeXchange via the PRIDE database (https://www.ebi.ac.uk/pride/) with the accession number: PXD054637.

The source data of this paper are collected in the following database record: biostudies:S-SCDT-10_1038-S44318-024-00333-9.

## Peer review information

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

## Acknowledgements

The study was financially supported by the Labex EpiGenMed, an Investissements d'avenir program, reference ANR-10-LABX-12-0, the Fondation pour la Recherche Médicale (FDT202012010454) awarded to LB and the ANR-AAPG2021-PROCONUC. We are grateful to the imaging facility MRI, a member of the national infrastructure France-BioImaging infrastructure supported by the French National Research Agency (ANR-10-INBS-04, Investments for the future). We also thank the Huib Ovaa laboratory for providing the ABPs and Dr Sophie Pattingre for providing the ATG7 knockout MEFs.

## Author contributions

**Lorène Brunello**: Conceptualization; Data curation; Investigation; Methodology. **Jolanta Polanowska**: Conceptualization; Data curation; Supervision; Investigation. **Léo Le Tareau**: Data curation; Investigation. **Chantal Maghames**: Data curation; Investigation. **Virginie Georget**: Data curation; Investigation; Methodology. **Charlotte Guette**: Data curation; Investigation; Methodology. **Karima Chaoui**: Data curation; Investigation. **Stéphanie Balor**: Data curation; Investigation. **Marie-Françoise O'Donohue**: Supervision; Writing—original draft. **Marie-Pierre Bousquet**: Data curation; Supervision. **Pierre-Emmanuel Gleizes**: Supervision; Writing—original draft; Writing—review and editing. **Dimitris P Xirodimas**: Conceptualization; Supervision; Funding acquisition; Writing—original draft; Project administration; Writing—review and editing.

Source data underlying figure panels in this paper may have individual authorship assigned. Where available, figure panel/source data authorship is listed in the following database record: biostudies:S-SCDT-10_1038-S44318-024-00333-9.

## Disclosure and competing interests statement

The authors declare no competing interests.

