## [Peer Review File · The EMBO Journal]

A nuclear protein quality control system for elimination of nucleolus-related inclusions

Lorene Brunello, Jolanta Polanowska, Leo Le Tareau, Chantal Maghames, Virginie Georget, Charlotte Guette, Karima Chaoui, Stephanie Balor, Marie O'Donohue, Marie-Pierre Bousquet, Pierre-Emmanuel Gleizes, and Dimitris Xirodimas

Corresponding author(s): Dimitris Xirodimas (dimitris.xirodimas@crbm.cnrs.fr)

Review Timeline:

Submission Date:	7th Feb 24
Editorial Decision:	22nd Mar 24
Revision Received:	19th Jul 24
Editorial Decision:	4th Sep 24
Revision Received:	18th Nov 24
Accepted:	26th Nov 24

Editor: Hartmut Vodermaier

Transaction Report:

Dr. Dimitris Xirodimas
CNRS
Centre de Recherche de Biochimie Macromoléculaire
Montpellier 34000
France

22nd Mar 2024

Re: EMBOJ-2024-116906
A nuclear Protein Quality Control system for the elimination of nucleolus-related inclusions

Dear Dr. Xirodimas,

Thank you for submitting your manuscript on nuclear protein quality control to The EMBO Journal. Three expert referees have now evaluated it and returned the below-copied reports. As you will see, the referees appreciate the importance of the topic and the potential interest of your main findings. At the same time, they remain (to varying degrees) unconvinced that the presented analyses have already led to sufficiently compelling understanding of the underlying mechanisms. Key issues are the over-reliance on inhibitors and on microscopic analyses, necessitating complementary genetic/biochemical approaches, and the uncertainty whether the effect of IGS42 is specific to this lncRNA. Should you be able to satisfactorily address these shared major concerns, as well as the various more specific queries, we would be happy to pursue a revised version of the manuscript further for publication.

Since it is our policy to aim for a single round of major revision, it is important to diligently respond to each referee point at the time of resubmission, and I would therefore encourage you to contact me with a preliminary point-by-point response already during the early stages of your revision work, in order to clarify how key issues may be solved and for us to agree on a revision plan. We would also be open to extension of the default three-months revision period if needed; our 'scooping protection' (meaning that competing work appearing elsewhere in the meantime will not affect our considerations of your study) would of course remain valid also throughout such an extension.

Further information on preparing, formatting and uploading a revised manuscript can be found below and in our Guide to Authors. Thank you again for the opportunity to consider this work for The EMBO Journal, and I look forward to hearing from you in due time.

Yours sincerely,

Hartmut Vodermaier

3) Revised manuscript text (including main tables, and figure legends for main and EV figures) has to be submitted as editable

text file (e.g., .docx format). We encourage highlighting of changes (e.g., via text color) for the referees' reference.

4) Each main and each Expanded View (EV) figure should be uploaded as individual production-quality files (preferably in .eps, .tif, .jpg formats). For suggestions on figure preparation/layout, please refer to our Figure Preparation Guidelines:

8) Please note that supplementary information at EMBO Press has been superseded by the 'Expanded View' for inclusion of additional figures, tables, movies or datasets; with up to five EV Figures being typeset and directly accessible in the HTML version of the article. For details and guidance, please refer to:

embopress.org/page/journal/14602075/authorguide#expandedview

9) Digital image enhancement is acceptable practice, as long as it accurately represents the original data and conforms to community standards. If a figure has been subjected to significant electronic manipulation, this must be clearly noted in the figure legend and/or the 'Materials and Methods' section. The editors reserve the right to request original versions of figures and the original images that were used to assemble the figure. Finally, we generally encourage uploading of numerical as well as gel/blot image source data; for details see: embopress.org/page/journal/14602075/authorguide#sourcedata

At EMBO Press, we ask authors to provide source data for the main manuscript figures. Our source data coordinator will contact you to discuss which figure panels we would need source data for and will also provide you with helpful tips on how to upload and organize the files.

Further information is available in our Guide For Authors:

In the interest of ensuring the conceptual advance provided by the work, we recommend submitting a revision within 3 months (20th Jun 2024). Please discuss the revision progress ahead of this time with the editor if you require more time to complete the revisions. Use the link below to submit your revision:

Link Not Available

Referee #1:

Protein aggregation is the hallmark of many neurodegenerative diseases. These aggregates are most often found in the cytoplasm, but aggregates in the nucleus have also been observed in some neurodegenerative diseases, like ALS and some polyQ expansion diseases. Understanding the mechanisms by which the formation of these aggregates can be prevented, or if aggregates have already formed can be reversed, is an important step in the development of treatments for these diseases. In their manuscript "A nuclear Protein Quality Control system for the elimination of nucleolus-related inclusions" Brunello and colleagues describe a part of the protein quality control network that is involved in the clearance of stress induced nuclear inclusions formed by the 60S ribosomal protein L11. The authors report that clearance of nuclear RPL11-GFP inclusions is not prevented by inhibition of autophagy or nuclear export but can be inhibited by inhibition of ubiquitination events that involve the E3 ligase HUWE1 and the proteasome. In the final part of the manuscript the authors also describe that Actinomycin D treatment and knockdown of the long noncoding RNA IGS42 prevents the clearance of aggregates.

This is not the first paper that describes the formation and clearance of stress induced inclusions associated with nucleoli, but it adds to our understanding of these processes by using a substrate with a complex pattern of posttranslational modifications (RPL11) and uses some novel techniques to analyze the clearance of inclusions (e.g. expansion microscopy, photoconversion experiments).

The two main new findings of this manuscript are the identification of HUWE1, an E3 ubiquitin ligase and of the lncRNA IGS42 as factors that both play a role in the clearance of RPL11 inclusions. Both findings are potentially interesting, but both are not followed up in detail, especially not regarding the question if these are general factors involved in quality control, or if they are

specific for RPL11 or in case of HUWE1 nuclear aggregates.

As mentioned above there are other publications describing the formation and clearance of nuclear inclusions in that also investigate the effects of Act D and Ver-155008 during clearance, or use proteasome inhibition to cause aggregation inside the nucleus (e.g. Azkanaz et al. 2019, Frottin et al. 2019, Mediani et al 2019). It is therefore not always clear in this manuscript what new insights can be gained from the experiments presented here, and it seems that the level of conceptual advance presented here is largely incremental, mainly supporting already published findings in slightly different experimental set ups. The advantages of the experiments used in this manuscript need to be explained and the differences between the experiments shown here, and those published in the past must be discussed in much more detail. Where results are contradictory it would be helpful to perform the different experiments site by site, and control for factors that might explain these differences (e.g. effects caused by different substrate of the inclusion or differences of Act D concentration).

The main weakness of this manuscript is however that it is purely descriptive and does not provide any mechanistic insights, it is therefore in its current form not suitable for publication in EMBO.

Other points:

- There seems to be an imbalance between the points highlighted in the abstract and the experiments included in the main Figures. The abstract mentions a role of CRM mediated nuclear export and that the observed effects are independent of autophagy/lysosome, these points are however not part of the main figures, whereas several panels contain data regarding Nedd8, that is not mentioned in the abstract, and which function is barely discussed. Nedd8 has as far as I can tell no influence on the findings discussed here, so it is not clear why it is included.
- It is conceptually not clear to me how the role of proteasome inhibition as a method to induce protein inclusions can be separated from a role during the clearance of aggregates, can the authors include controls to distinguish these possibilities?
- Controls using different stress induced aggregates in the nucleus but also the cytoplasm are missing, in the absence of these it is difficult to answer the question if any of the described effects are specific to inclusions associated with nucleoli.
- In Figure 2 it does not become clear how areas A and B are defined, what markers were used to distinguish these areas, and what they represent.
- Not all figures contain scale bars (e.g. Figure 2 A and D)
- For Figure 8 clear criteria what defines a ubiquitin positive inclusion are necessary, for me the structures in the IGS42 siRNA panels MG132 vs recovery look very different and are not comparable.
- To my understanding the RPL11-GFP panels should look comparable between Figure 9 A and B, that however does not seem to be the case, can the authors explain these differences.

Referee #2:

In the presented manuscript, Brunello and colleagues explored the cellular system capable of removing misfolded proteins accumulated in the cell nucleus, particularly in stress-induced (MG-132-induced) nucleolus-related inclusions. The authors focused on characterizing the elimination process of nucleolar inclusions during the recovery period. They observed that this process is independent of autophagy and does not occur in the cytoplasm. Furthermore, the process of removal of nucleolar inclusions requires the Ubiquitin-activating enzyme E1 as well as E3 ubiquitin-protein ligase HUWE1. Interestingly, the recovery process is functionally linked to the synthesis of lncRNA IGS42 transcribed from the intergenic spacer by dedicated RNA polymerase I. This is an interesting manuscript focused on the less understood mechanism of cellular recovery after the stress-related removal of damaged or misfolded proteins, which are polyubiquitinated and directed to the 20S proteasome for subsequent processive degradation. Since this work largely depends on specific inhibitors, which have additional side effects, some interpretations need more mechanistic insight.

Here are my major points:

The authors used only one marker, ribosomal protein L11 (tagged with GFP), as a reporter for the functional study. This protein is an essential responder to nucleolar stress, but other ribosomal proteins, including RPL5, RPL23, and RPS7, also respond to this stress challenge. Therefore, the authors should use at least one additional ribosomal protein to test their hypothesis.

To study the role of the nucleolar transcription of ribosomal genes in removing stress-induced nucleolus-related inclusions, the authors used the specific RNA pol I inhibitor Actinomycin D (ActD), which they removed from cells for cell recovery. ActD is a DNA intercalator that causes massive chromatin condensation, including ribosomal genes. I doubt the authors do not see any significant nucleolar rearrangement after one hour of ActD treatment. Also, the complete removal of ActD from the cells for recovery is not clear. Furthermore, the authors used another RNA pol I inhibitor, CX-5461, as a control, which irreversibly blocks RNA pol I, causes DNA damage, and disrupts cell viability. Therefore, the conclusions about the role of endogenous transcription of ribosomal genes in removing nucleolar inclusions based on these two inhibitors with possible side effects are at least questionable. Therefore, alternative small molecule inhibitors of RNA pol I transcription (BMH-21) should be used.

The manuscript almost exclusively depends on the microscopy approaches. The authors should isolate the cytoplasmic fraction and nucleoli in the stress-recovery time-course experiment and perform quantitative western blot/(alternatively mass spec) comparative analysis on the levels of candidate misfolded proteins accumulated in nucleolar inclusions. The elimination of the misfolded candidate proteins in the nucleolar fractions should be quantified.

What is the effect of a more specific proteasome inhibitor, Epoxomicin, on the formation of nucleolar inclusions?

The exclusion of unstable processing intermediates of pre-rRNA processing, ITS1, and ITS2, by RNA FISH presented in Figure 9B could be better. The authors should also visualize stable, mature 18S and 28 rRNAs. They should use higher magnification and preferentially super-resolution microscopy.

The authors should check the expression levels of candidate misfolded proteins in the nucleolar inclusions before and in the recovery time series by qRT-PCR. They should correlate it with protein degradation.

Referee #3:

In this study, Brunello and co-workers show that the nuclear ubiquitin-proteasome system is responsible for the clearance of nuclear inclusions. Consistent with earlier studies, they found that nucleoli play an important role in sequestration of misfolded proteins and in the formation of intranuclear inclusions. Probably the most intriguing finding of the study is that the authors also deciphered a mechanism that ensures that nuclear inclusions are amendable for clearance after stress and in doing so prevent the formation of liquid-solid phase separated inclusions. An RNA pol I produced long non-coding RNA, IGS42, appears to play a critical role in this process. This is a thorough study that provides important new insights into the role of nuclear protein quality control in preventing the formation of insoluble inclusions. Given the important role of nuclear inclusions in neurodegenerative diseases and the relatively limited insights in handling of these structures (as opposed to cytosolic inclusions), this study is likely to attract broad interest from the scientific community.

Comments

Fig 6C. The percentage of cells that have inclusions in the presence of ActD are also reduced during the recovery. The difference before and after stress looks similar in magnitude from what is observed in the absence of ActD. It is admittedly non-significant in the presence of ActD but this seems to be largely due to a larger variation in this particular condition. It is important to know for the interpretation if there is a difference as an alternative explanation would be that ActD treatment increases the formation of inclusions without substantially affecting the clearance. More may be left after the recovery because there were more to start from. More data points could clarify this issue.

Fig 7. There are several other lncRNAs that show a similar pattern even though it is less dramatic than IGS42 (IGS24, IGS27.5, IGS28). To substantiate the claim that IGS42 is specifically responsible for the effect on inclusions clearance it would be important to check if depletion of other lncRNA(s) can have a similar effect.

Minor comments

Fig 5A. The schematic drawing showing the effects of the UBAi and NAEi on Neddylation is helpful but the text in the result section is less clear. It could be more explicitly mentioned what the inhibitors do and do not inhibit.

The abbreviation "RP" is not introduced in the manuscript.

We would like to thank the referees for their time in reading our manuscript and for their constructive comments that we feel helped in improving our manuscript. In the revised version the new added information/changes are highlighted in red whereas in yellow the new concepts/insights of the study and advantages of the used methodology compared to that used in previous studies.

Referee #1:

-Protein aggregation is the hallmark of many neurodegenerative diseases. These aggregates are most often found in the cytoplasm, but aggregates in the nucleus have also been observed in some neurodegenerative diseases, like ALS and some polyQ expansion diseases. Understanding the mechanisms by which the formation of these aggregates can be prevented, or if aggregates have already formed can be reversed, is an important step in the development of treatments for these diseases.

In their manuscript "A nuclear Protein Quality Control system for the elimination of nucleolus-related inclusions" Brunello and colleagues describe a part of the protein quality control network that is involved in the clearance of stress induced nuclear inclusions formed by the 60S ribosomal protein L11. The authors report that clearance of nuclear RPL11-GFP inclusions is not prevented by inhibition of autophagy or nuclear export but can be inhibited by inhibition of ubiquitination events that involve the E3 ligase HUWE1 and the proteasome. In the final part of the manuscript the authors also describe that Actinomycin D treatment and knockdown of the long noncoding RNA IGS42 prevents the clearance of aggregates. This is not the first paper that describes the formation and clearance of stress induced inclusions associated with nucleoli, but it adds to our understanding of these processes by using a substrate with a complex pattern of posttranslational modifications (RPL11) and uses some novel techniques to analyze the clearance of inclusions (e.g. expansion microscopy, photoconversion experiments).

The two main new findings of this manuscript are the identification of HUWE1, an E3 ubiquitin ligase and of the lncRNA IGS42 as factors that both play a role in the clearance of RPL11 inclusions. Both findings are potentially interesting, but both are not followed up in detail, especially not regarding the question if these are general factors involved in quality control, or if they are specific for RPL11 or in case of HUWE1 nuclear aggregates.

Response

The new data on the biochemical characterization of ribosomal proteins (RPL11, RPL7, RPS3) in the cytoplasm or in nuclear inclusions, shows that the observed response during stress is specific for the nuclear inclusions (Fig. 3). The proteomics analysis now shows that the stress response characterised in detail by imaging analysis is not unique to RPL11 (the main model substrate), but many RPs collectively display a similar phenotype, i.e. accumulate in inclusions upon stress and are eliminated during the recovery period (Fig. 3). Additionally, HUWE1 is shown both by proteomics and by imaging analysis to accumulate in nuclear inclusions upon stress. While it is still not clear whether HUWE1 specifically targets RPs or also controls the modification/stability of other proteins within the formed inclusions, the study now defines that the key role of HUWE1 and the Ubiquitin system is not during inclusion formation but rather during the elimination process. Despite the fact that multiple E3-ligases maybe involved in the stress response, the observation that HUWE1 knockdown completely blocks inclusion elimination, mimicking the effect of the Ubiquitin E1 inhibitor, shows that HUWE1 is the key E3-ligase for the response. We believe these are critical new

findings for the field as generally, the role of the Ubiquitin/Ubl system in protein aggregation remains elusive.

-As mentioned above there are other publications describing the formation and clearance of nuclear inclusions in that also investigate the effects of Act D and Ver-155008 during clearance, or use proteasome inhibition to cause aggregation inside the nucleus (e.g. Azkanaz et al. 2019, Frottin et al. 2019, Mediani et al 2019). It is therefore not always clear in this manuscript what new insights can be gained from the experiments presented here, and it seems that the level of conceptual advance presented here is largely incremental, mainly supporting already published findings in slightly different experimental set ups.

Response

To our knowledge the above-mentioned studies have mainly focused on the formation of nuclear inclusions, whereas the elimination process is currently poorly characterized. In particular, whether proteins within these inclusions are refolded back into an active state or are targeted for elimination during the recovery period remains unclear. Additionally, the role of the Ubiquitin system in aggregate formation/elimination also remains poorly defined. More specifically, key missing information include the definition of the required elements for the recovery process and the detailed characterization of changes in the biophysical properties of nuclear inclusions during the stress period. Our findings on the later aspect are conceptually important as they now provide a model for the fate of misfolded proteins during recovery: Depending on the extent of stress and the subsequent changes in the biophysical properties of the formed inclusions (transition from liquid to solid phase), misfolded proteins can either refold or targeted for elimination. This now proposes a unifying model for previous and the presented study. We believe our study provides a significant advance on the above-described aspects and we have now emphasized these findings in the manuscript (Introduction, Discussion).

-The advantages of the experiments used in this manuscript need to be explained and the differences between the experiments shown here, and those published in the past must be discussed in much more detail. Where results are contradictory it would be helpful to perform the different experiments site by site, and control for factors that might explain these differences (e.g. effects caused by different substrate of the inclusion or differences of Act D concentration).

Response

In the revised version we addressed the above issues by clearly stating the advantages/differences between the experiments performed here with those performed in previous studies and the new introduced insights/concepts. In particular, we highlight the use of expansion microscopy, CLEM, 3D-RIM, photoconversion that compared to previous studies they now provide detailed imaging analysis on the re-organisation of the nucleolus upon stress (both at the protein and RNA level) and critically, determined the fate of proteins found in aggregates during the recovery period (photoconversion). These paragraphs are now highlighted in yellow.

-The main weakness of this manuscript is however that it is purely descriptive and does not provide any mechanistic insights, it is therefore in its current form not suitable for publication in EMBO.

Response

While in principle we agree with this comment we also find it a bit harsh. We are convinced the referee is aware that the vast majority of articles in this area of research are mainly based on imaging analysis and characterization of morphological changes in the nucleolus.

Examples of recently published articles in the field include:

-Frottin F, Schueder F, et al. (2019) The nucleolus functions as a phase-separated protein quality control compartment. *Science* 365: 342–347

-Ali A, Garde R, Schaffer OC, Bard JAM, Husain K, Kik SK, Davis KA, Luengo-Woods S, Igarashi MG, Drummond DA, et al (2023) Adaptive preservation of orphan ribosomal proteins in chaperone-dispersed condensates. *Nat Cell Biol.* 11:1691-1703.

We believe our article advances our understanding both on how the nucleolus responds to stress and critically to identify new key elements for the elimination process, which is poorly characterized. Additionally, the inclusion of new data, especially the SILAC-based proteomics provide a biochemical basis on aggregate formation/elimination that fully supports the data derived from imaging analysis.

Other points:

- There seems to be an imbalance between the points highlighted in the abstract and the experiments included in the main Figures. The abstract mentions a role of CRM mediated nuclear export and that the observed effects are independent of autophagy/lysosome, these points are however not part of the main figures, whereas several panels contain data regarding Nedd8, that is not mentioned in the abstract, and which function is barely discussed. Nedd8 has as far as I can tell no influence on the findings discussed here, so it is not clear why it is included.

Response

We opted to include the autophagy/nuclear export data as supplementary figures due to their large number and as we found no role of these pathways in the elimination process. However, as they are part of supplementary material but still new (not complementary) information, we included a statement in the abstract. Regarding the role of NEDD8, we agree that we have not fully dissected its role on the response and thus we did not include a statement in the abstract. However, we used it as marker of the proteotoxic stress response, especially for the recovery period (Fig. 3). As we describe in the text, NEDD8 conjugation on proteins can be achieved through 2 pathways: The so-called “canonical” which mainly involves NEDDylation of cullins and the control of cullin-ring E3-ligases and the “atypical” which is mainly observed upon proteotoxic stress conditions. It is characterized by the use of the Ubiquitin system and the formation of hybrid NEDD8-Ubiquitin chains. In Fig. 5 we found that the recovery process strictly depends on HUWE1 E3-ligase that specifically promotes atypical but not canonical NEDDylation (Maghames et al., 2018, Nat. Com.).

Currently, the collective use of inhibitors of the Ubiquitin E1, NEDD8 E1 (canonical) enzymes and HUWE1 inhibition represents the best available approach to assess the role of atypical NEDDylation in the aggregation process. We think this is important for the field for the following reasons: 1. In contrast to the well-established function of canonical NEDDylation, our knowledge on atypical NEDDylation is rather limited, with few reports on aggregate formation. 2. The data on NEDD8/HUWE1 indicate potential mechanisms (e.g. the formation of hybrid NEDD8-Ubiquitin conjugates) through which E3-ligases in general and particularly HUWE1 controls aggregate elimination. These points are only included in the discussion as possibilities and thus were not included in the abstract. We feel that this aspect

of NEDD8 biology (atypical NEDDylation), based on the activity of HUWE1, should at least be discussed as a potential regulatory mechanism of aggregate dynamics.

- It is conceptually not clear to me how the role of proteasome inhibition as a method to induce protein inclusions can be separated from a role during the clearance of aggregates, can the authors include controls to distinguish these possibilities?

Response

We have used previously published concentrations of MG132 (0.5 μ M) that while they reduce proteasome activity (confirmed in S4), they do not induce proteotoxic stress. This was evaluated by the absence of increase in NEDD8 conjugates (as marker of proteotoxicity, S2A), lack of aggregate formation (Fig. 4A). This is regarded as a classic approach to assess the role of the proteasome in aggregate dynamics and described in page 7. Based on the observation that the used low doses of MG132 (0.5 μ M) added only during the recovery period completely inhibit the elimination process, we propose that full proteasome activity is required for aggregate elimination. This indicates that proteasome activity is a limiting factor in the recovery process. The new data on the use of irreversible proteasome inhibitors (Epoxomicin) further enhance this notion, as in this case, no aggregate elimination is observed during the recovery period.

- Controls using different stress induced aggregates in the nucleus but also the cytoplasm are missing, in the absence of these it is difficult to answer the question if any of the described effects are specific to inclusions associated with nucleoli.

Response

We have used two type of stress conditions, mainly proteasome inhibition by MG132 but also heat shock. MG132 is known to induce aggregates both in the cytoplasm (aggresomes) and in the nucleus (nucleolus-related inclusions), while heat shock mainly affects nucleolar morphology. To our knowledge these are the main stress conditions commonly used to induce aggregation and were both used in this study.

We have now followed biochemical approaches, including proteome-wide characterization of aggregates from nuclear enriched fractions and western blot analysis from either cytoplasmic fractions or fractions enriched for inclusions. For the chosen model substrates, the data show that the recovery process is specific for nuclear inclusions, not observed in the cytoplasm (Fig. 3J).

- In Figure 2 it does not become clear how areas A and B are defined, what markers were used to distinguish these areas, and what they represent.

Response

RPL11 staining was used to define areas A and B. In unstressed conditions RPL11 provides a rather homogenous staining in the nucleolus (Fig. 2D ctr). Upon treatment with MG132 and nucleolar re-organisation, RPL11 appears either in “bright” fluorescent aggregates (state B) or outside of this defined area and is regarded as state A. Thus, RPL11 was a good model nucleolar protein to monitor the biophysical properties of these two states by FRAP. The clear difference in RPL11 motility confirms the presence of these two states that are induced within the nucleolus upon stress. These details are now included in the main text (page 6).

- Not all figures contain scale bars (e.g. Figure 2 A and D)

Response

We have now included scale bars in Fig.2A, D.

- For Figure 8 clear criteria what defines a ubiquitin positive inclusion are necessary, for me the structures in the IGS42 siRNA panels MG132 vs recovery look very different and are not comparable.

Response

The “ring-like” structures stained with Ubiquitin that specifically appear upon stress in the nucleus is the main feature applied throughout the study. We have established that these “ring-like” structures are the nucleolus-related inclusions that contain ribosomal proteins. In some cases, the stress-induced separation of RPL11 and fibrillarin staining was used. We have now replaced the IGS₄₂ siRNA MG132 figure in recovery.

- To my understanding the RPL11-GFP panels should look comparable between Figure 9 A and B, that however does not seem to be the case, can the authors explain these differences.

Response

We do not feel there was significant difference in RPL11 staining between the two panels, maybe in 9B the intensity in the control panel was lower compared to 9A. We have now replaced this set of figures.

Referee #2:

-In the presented manuscript, Brunello and colleagues explored the cellular system capable of removing misfolded proteins accumulated in the cell nucleus, particularly in stress-induced (MG-132-induced) nucleolus-related inclusions. The authors focused on characterizing the elimination process of nucleolar inclusions during the recovery period. They observed that this process is independent of autophagy and does not occur in the cytoplasm. Furthermore, the process of removal of nucleolar inclusions requires the Ubiquitin-activating enzyme E1 as well as E3 ubiquitin-protein ligase HUWE1. Interestingly, the recovery process is functionally linked to the synthesis of lncRNA IGS42 transcribed from the intergenic spacer by dedicated RNA polymerase I. This is an interesting manuscript focused on the less understood mechanism of cellular recovery after the stress-related removal of damaged or misfolded proteins, which are polyubiquitinated and directed to the 20S proteasome for subsequent processive degradation. Since this work largely depends on specific inhibitors, which have additional side effects, some interpretations need more mechanistic insight.

Here are my major points:

-The authors used only one marker, ribosomal protein L11 (tagged with GFP), as a reporter for the functional study. This protein is an essential responder to nucleolar stress, but other ribosomal proteins, including RPL5, RPL23, and RPS7, also respond to this stress challenge. Therefore, the authors should use at least one additional ribosomal protein to test their hypothesis.

Response

In addition to RPL11, we have now used RPL7 and RPS7 as model ribosomal proteins (available constructs in our laboratory). Similarly to RPL11 they accumulate in the stress induced nucleolus-related inclusions marked with Ubiquitin. Additionally, the new SILAC-

based proteomics analysis shows that ribosomal proteins collectively (>40) accumulate in stress-induced aggregates. We believe the proteomics data provide strong biochemical evidence to support the hypothesis that ribosomal proteins accumulate in stress induced aggregates.

-To study the role of the nucleolar transcription of ribosomal genes in removing stress-induced nucleolus-related inclusions, the authors used the specific RNA pol I inhibitor Actinomycin D (ActD), which they removed from cells for cell recovery. ActD is a DNA intercalator that causes massive chromatin condensation, including ribosomal genes. I doubt the authors do not see any significant nucleolar rearrangement after one hour of ActD treatment. Also, the complete removal of ActD from the cells for recovery is not clear. Furthermore, the authors used another RNA pol I inhibitor, CX-5461, as a control, which irreversibly blocks RNA pol I, causes DNA damage, and disrupts cell viability. Therefore, the conclusions about the role of endogenous transcription of ribosomal genes in removing nucleolar inclusions based on these two inhibitors with possible side effects are at least questionable. Therefore, alternative small molecule inhibitors of RNA pol I transcription (BMH-21) should be used.

Response

We used the BMH-21 inhibitor and we obtained very similar results to the use of ActD or CX-5461, i.e. that pre-treatment with BMH-21 blocks the elimination of stress-induced inclusions (S6B). However, we do not present data on the addition of BMH-21 only during the recovery period of 8h, as we found increased cell toxicity compared to the use of ActD or CX-5461 that did not induce toxicity during the same period.

However, while we agree with the raised concerns regarding the use of ActD/CX-5461, we want to emphasize that we used very low doses of ActD, (5nM) and performed kinetic analysis to indeed find conditions where no or minimal morphological changes in the nucleolus were observed. Additionally, the Pol I inhibitors were used both prior to stress (for 1hr) and during the whole recovery period (8h). However, we only see inhibition in aggregate elimination if the inhibitors are added prior to stress and not if added during the recovery period. We therefore believe that the observations cannot be simply explained due to the DNA damage response that is induced by the inhibitors. Finally, the use of Pol I inhibitors was the initial/guiding step in characterizing the role of Pol I in aggregate elimination. The discovery and characterization of the lncRNA IGS₄₂ is the key novel part of the manuscript.

-The manuscript almost exclusively depends on the microscopy approaches. The authors should isolate the cytoplasmic fraction and nucleoli in the stress-recovery time-course experiment and perform quantitative western blot/(alternatively mass spec) comparative analysis on the levels of candidate misfolded proteins accumulated in nucleolar inclusions. The elimination of the misfolded candidate proteins in the nucleolar fractions should be quantified.

Response

We decided to use ribosomal proteins as model substrates for the biochemical characterization, mainly because our imaging analysis is focused on ribosomal proteins and because they exist both in the cytoplasm and in the nucleolus. We isolated cytoplasmic fractions and nucleolus-related aggregates from unstressed, stressed and recovery period cells. By western blot analysis we see a clear enrichment of ribosomal proteins in the isolated nucleolus-related aggregates which is reversed during the recovery period, indicative of the

elimination process. In contrast, under the same conditions no effect in ribosomal protein levels was observed in the cytoplasm.

Additionally, we performed SILAC based quantitative proteomics on isolated aggregates upon stress and recovery period. We followed protocols that enrich for nuclear fractions. We acknowledge that it is impossible to exclude that some cytoplasmic inclusions are also co-purified. However, based on the analysis of the proteomics data mainly for ribosomal proteins, Ubiquitin and HUWE1, we see a very similar profile to that observed by imaging analysis on nuclear aggregates. The key conclusion of the proteomic analysis is that ribosomal proteins collectively accumulate in the inclusions upon stress and are eliminated during the recovery period. We feel this now provides strong biochemical evidence that ribosomal proteins (and many other potential substrates) are eliminated from the nuclear inclusions during recovery, consistent with the imaging analysis. This is a key conclusion of the study.

We also believe that the proteomics analysis will be helpful for the research community working in protein aggregation. To our knowledge this is the first proteomics analysis of nuclear aggregate composition during the recovery period.

In addition, the observation that nuclear aggregate elimination is independent of the autophagy system, which is essential for cytoplasmic aggregate elimination, further supports the notion for the existence of protein quality control pathways that specifically operate in distinct compartments.

-What is the effect of a more specific proteasome inhibitor, Epoxomicin, on the formation of nucleolar inclusions?

Response

We have now used Epoxomicin and similarly to MG132, Epoxomicin induces the formation of nucleolus-related inclusions. However, as Epoxomicin is an irreversible inhibitor (in contrast to MG132 which is reversible), we see no aggregate elimination during the recovery period (S4). We believe this further emphasizes the critical role of the proteasome activity during recovery for aggregate elimination.

-The exclusion of unstable processing intermediates of pre-rRNA processing, ITS1, and ITS2, by RNA FISH presented in Figure 9B could be better. The authors should also visualize stable, mature 18S and 28 rRNAs. They should use higher magnification and preferentially super-resolution microscopy.

Response

FISH was performed with a probe hybridizing with the mature 18S rRNA sequence. Unlike with the probes to the transcribed spacers, the cytoplasm was strongly labelled, reflecting the massive amount of mature ribosomes. This makes it more difficult to detect the nucleolar signal, but some labelling was detected in the nucleoli. Critically, the observed labelling was excluded from the RPL11-eGFP labelled inclusions, confirming the observations made with the pre-rRNAs probes (S8).

We now performed 3D random illumination microscopy (3D-RIM) to provide high-resolution images for ITS1/2 (Fig. 9).

-The authors should check the expression levels of candidate misfolded proteins in the nucleolar inclusions before and in the recovery time series by qRT-PCR. They should correlate it with protein degradation.

Response

We performed qRT-PCR for RPL11, RPL7, RPS7 and did not observe differences in gene expression either upon stress or during the recovery period relatively to the unstressed conditions (S2H).

Referee #3:

-In this study, Brunello and co-workers show that the nuclear ubiquitin-proteasome system is responsible for the clearance of nuclear inclusions. Consistent with earlier studies, they found that nucleoli play an important role in sequestration of misfolded proteins and in the formation of intranuclear inclusions. Probably the most intriguing finding of the study is that the authors also deciphered a mechanism that ensures that nuclear inclusions are amendable for clearance after stress and in doing so prevent the formation of liquid-solid phase separated inclusions. An RNA pol I produced long non-coding RNA, IGS42, appears to play a critical role in this process. This is a thorough study that provides important new insights into the role of nuclear protein quality control in preventing the formation of insoluble inclusions. Given the important role of nuclear inclusions in neurodegenerative diseases and the relatively limited insights in handling of these structures (as opposed to cytosolic inclusions), this study is likely to attract broad interest from the scientific community.

Comments

-Fig 6C. The percentage of cells that have inclusions in the presence of ActD are also reduced during the recovery. The difference before and after stress looks similar in magnitude from what is observed in the absence of ActD. It is admittedly non-significant in the presence of ActD but this seems to be largely due to a larger variation in this particular condition. It is important to know for the interpretation if there is a difference as an alternative explanation would be that ActD treatment increases the formation of inclusions without substantially affecting the clearance. More may be left after the recovery because there were more to start from. More data points could clarify this issue.

Response

We agree in principle with the referee's point. We now indicate the values for each condition and while we see almost 60% recovery in MG132 treated cells (42.3 vs 18.9), we only see 20% recovery in ActD treated cells (86.5 vs 69.7). Even if ActD indeed increases the % of cells with aggregates, we think it is clear that the recovery process is severely compromised based on the % of cells with remaining aggregates. Similar observations were obtained with the use of CX-5461 and BMH-21 Pol I inhibitors. We would like to point out that the use of Pol I inhibitors was the initial/guiding step in characterizing the role of Pol I in aggregate elimination. The discovery and characterization of lncRNA IGS₄₂ is the key novel part of the manuscript.

-Fig 7. There are several other lncRNAs that show a similar pattern even though it is less dramatic than IGS42 (IGS24, IGS27.5, IGS28). To substantiate the claim that IGS42 is specifically responsible for the effect on inclusions clearance it would be important to check if depletion of other lncRNA(s) can have a similar effect.

Response

We have now tested the effect of knockdown of a series of additional IGS lncRNAs (IGS16, 18, 20) that did not fulfill the criteria of selection, i.e. were either not induced by MG132 and/or their expression does not depend on RNA Pol I. Additionally, IGS16 and the IGS20 locus were previously shown to participate in aggregate formation for a different stress response (heat shock). The selection of tested IGS lncRNAs was also based on the efficiency of knockdown. For the above-described reasons we believe such lncRNAs represent appropriate specificity controls for IGS42. Here, we see no effect on the recovery process, indicating specific roles for these lncRNAs (such as IGS42) in the stress response and aggregate elimination. Consistent with previous studies the role of these lncRNAs in aggregate dynamics may be stress-dependent.

Minor comments

-Fig 5A. The schematic drawing showing the effects of the UBAi and NAEi on Neddylation is helpful but the text in the result section is less clear. It could be more explicitly mentioned what the inhibitors do and do not inhibit.

Response

We have now added a paragraph to explain in more detail the 2 modes of protein NEDDylation, canonical vs atypical (page 8).

-The abbreviation "RP" is not introduced in the manuscript.

Response

We have now introduced the "RP" abbreviation.

Dr. Dimitris Xirodimas
CNRS
Centre de Recherche de Biochimie Macromoléculaire
Montpellier 34000
France

4th Sep 2024

Re: EMBOJ-2024-116906R
A nuclear Protein Quality Control system for the elimination of nucleolus-related inclusions

Dear Dr. Xirodimas,

Thank you for submitting your revised manuscript to The EMBO Journal. It has now be re-reviewed by the three original referees, whose reports are copied below. As you will see, all referees acknowledge the improvements to the manuscript. However, each one of them retains specific major concerns that continue to preclude publication at this stage, especially where the issues that had already been raised during the initial round of review.

Given that many other points appear to have been addressed to the referees' satisfaction and that the reviewers continue to express interest, I would in this case allow an exceptional second round of major revision, so that you can deal with the remaining issues. But I need to stress that this will be the final round of revision, and that it will be essential to substantially strengthen IGS42 detection by RNA FISH (referee 2) and to include tests of the three stress-inducible/Poll-dependent lncRNAs IGS24, IGS27.5, IGS28 that referee 3 had already requested initially. In addition, it will be important to still put the current analyses and results better in the context of the existing literature (referee 1), and to consider/discuss/exclude alternative interpretations of the data, also in light of (yet to-be-reviewed) findings reported by M. Zhou et al (taking referee 1's thoughts on this below into serious consideration).

Should you decide to decisively address these outstanding issue now and to resubmit a re-revised manuscript to The EMBO Journal, please make sure to also incorporate a number of remaining editorial issues:

- Our routine revision image checks indicate that the control panels have been duplicated between Figures 2A and 2D; while the legend only states that 2D shows a "similar experiment as in A". This needs to be clarified, and if justifiable, explicitly mentioned in the figure legend.
- in Figure 8B, the error bars remain to be defined.
- Please add a header/legend in Table EV1, within the file itself.
- Please carefully double-check the reference list to make sure all citations are complete (some are currently missing page numbers/elocators, e.g., Sung et al 2016).
- Please rename the Conflict of Interest section into "Disclosure and Competing Interests Statement", in accordance with our updated Guide to Authors (<https://www.embopress.org/competing-interests>)
- As we are switching from a free-text author contribution statement towards a more formal statement based on Contributor Role Taxonomy (CRediT) terms, please remove the present Author Contribution section and instead specify each author's contribution(s) directly in the Author Information page of our submission system during upload of the final manuscript. See <https://casrai.org/credit/> for more information.
- Please adjust the order of the manuscript sections: Title page with complete author information, Abstract, Keywords, Introduction, Results, Discussion, Methods, Data Availability, Acknowledgements, Disclosure and Competing Interests Statement, References, Figure Legends.
- Please include suggestions for a short 'blurb' text prefacing and summing up the conceptual aspect of the study in two sentences (max. 250 characters), followed by 3-5 one-sentence 'bullet points' with brief factual statements of key results of the paper; they will form the basis of an editor-written 'Synopsis' accompanying the online version of the article. Please also upload a synopsis image, which can be used as a "visual title" for the synopsis section of your paper. The image should be in PNG or JPG format, and please make sure that it remains in the modest dimensions of (exactly) 550 pixels wide and 300-600 pixels high.
- Finally, our routine text similarity checks found several passages in the Methods section being near-verbatim copies from

previous papers by yourselves or others (e.g., 3D-RIM, CLEM, MS analysis). While this is in principle acceptable in the interest of reproducibility, the respective earlier articles should however be referenced at the start of each copied passage (e.g., "xxx was done essentially as reported by yyy et al").

I am therefore returning the manuscript to you for a second, final round of experimental revision, with the link below for eventual resubmission. Should you have any questions regarding the referee comments or this decision, please do not hesitate to contact me directly.

Yours sincerely,

Hartmut Vodermaier

9) To facilitate reproducibility and cross-laboratory adoption of methodologies, please structure the Materials & Methods section as outlined in our guide to authors, including a completed Reagents and Tools Table that can be downloaded from our author guidelines as well (<https://www.embopress.org/page/journal/14602075/authorguide#structuredmethods>).

10) Digital image enhancement is acceptable practice, as long as it accurately represents the original data and conforms to community standards. If a figure has been subjected to significant electronic manipulation, this must be clearly noted in the figure legend and/or the 'Materials and Methods' section. The editors reserve the right to request original versions of figures and the

original images that were used to assemble the figure. Finally, we generally encourage uploading of numerical as well as gel/blot image source data; for details see: embopress.org/page/journal/14602075/authorguide#sourcedata

At EMBO Press, we ask authors to provide source data for the main manuscript figures. Our source data coordinator will contact you to discuss which figure panels we would need source data for and will also provide you with helpful tips on how to upload and organize the files.

In the interest of ensuring the conceptual advance provided by the work, we recommend submitting a revision within 3 months (3rd Dec 2024). Please discuss the revision progress ahead of this time with the editor if you require more time to complete the revisions. Use the link below to submit your revision:

Link Not Available

Referee #1:

I believe that the additional text to put these results into a wider context in the introduction and the discussion is helpful. However, I think that even more background information would be helpful, and acknowledging the current state is not diminishing the quality of these results, e.g. I believe that the suggestion of the authors that "the role of the Ubiquitin system in aggregate formation/elimination ... remains poorly defined", could use some additional discussion/references to make clear what is already known, e.g. the presence of an autophagy-independent mechanism of aggregate clearance in the nucleus that relies on Hsp70, ubiquitination and the 26S proteasome has already been reported previously (Hjerpe et al Cell 2016).

The authors convincingly show in their mass spec analysis of enriched nuclear fractions that other ribosomal proteins also follow the fate of RPL11. However I would have liked to see a somewhat more detailed discussion of the identified proteins, and the characterization of the two fractions analyzed. At the moment this experiment is not well connected to the rest of the manuscript, and it is very hard to understand the data in the supplemental table.

It however is still not clear to me how it can be concluded from these experiments that the described quality control pathway is specific for nuclear protein quality control. This is especially important since there are experiments described in a manuscript deposited in BioRxiv that suggest that HUWE1 plays a general role in the clearance of aggregated proteins and that its role is not limited to aggregates in the nucleus (doi: <https://doi.org/10.1101/2023.05.30.542866>). As far as I can tell none of the experiments described there are contradicting the results presented here for RPL11, but they should influence the interpretation of the results, since a role of HUWE1 in the clearance of aggregates in the cytosol would suggest that this is not a quality control mechanism specific for the nucleus.

I believe these points can be addressed by augmenting the text.

Minor points:

In the point-by-point response the authors state that: "Additionally, HUWE1 is shown both by proteomics and by imaging analysis to accumulate in nuclear inclusions upon stress." Can the authors point me towards the figure where the presence of HUWE1 is shown by "imaging analysis"?

The source data files seem to not always be annotated correctly e.g. the blots in Figure 3J seem to be in the folder for 3A.

Due to the complication of a non-peer reviewed finding that contradicts the main message of the manuscript as stated in the title, I feel not confident in judging the general significance and priority of this manuscript as well as the question if the conclusions of the paper are justified, and have decided to answer them with in the current version of the paper with "n/a". addressed by

Referee #2:

I want to thank the authors for responding to the comments and for their extensive experimental work to revise the manuscript. The authors stressed in the manuscript the significance of the recovery process of stress-induced nuclear inclusions, which requires the RNA Pol I-dependent production of the lncRNA IGS42. They claimed that lncRNA IGS42 localizes specifically within the formed inclusions and promotes their elimination by preserving the mobility of resident proteins. This important fact must be appropriately documented in the high-quality RNA FISH localization. I do not see it in Figure 9; the background is so

high. The authors must significantly improve the quality of the RNA FISH detection of IGS42.

Referee #3:

While I was (and remain) excited about the data presented in this study, I am somewhat puzzled by the response of the authors to my request to test if the observed effect is specific for IGS42 or also depends on other stress-induced, pol I-dependent lncRNAs. In my request, I even listed three lncRNAs (IGS24, IGS27.5, IGS28) that from their own analysis seem to match their two selection criteria: stress induction and pol I dependency. For reasons that I do not comprehend, the authors performed the experiment instead for three lncRNAs (IGS16, 18, 20) that did not match their original selection criteria. Since these lncRNAs are not stress inducible and/or independent of pol I, it seems beforehand unlikely that they will have the same effect on clearance of inclusions as IGS42, which was indeed the outcome of the experiments. With that the question remains if the ability to make nuclear inclusion amenable to nuclear clearance mechanisms is unique to IGS42 or also applies to other stress-inducible, pol I-dependent lncRNAs. The outcome of the experiments may not affect the novelty or impact of the study but is relevant information for researchers who want to study the mechanisms underlying this phenomenon in more depth. If this experiment cannot be performed for technical reasons, this should be mentioned and the authors should discuss the possibility that they may be dealing with a more general feature of this class of lncRNAs in contrast to a unique property for IGS42.

We would like to thank the referees for their positive assessment of our revised version and for the additional comments. We have now responded to their remaining concerns. The new added information is highlighted in yellow in the manuscript.

Editorial Issues

- *Our routine revision image checks indicate that the control panels have been duplicated between Figures 2A and 2D; while the legend only states that 2D shows a "similar experiment as in A". This needs to be clarified, and if justifiable, explicitly mentioned in the figure legend.*

Response

We now provide the following statement regarding the experiment in Fig. 2A/2D. The experiments in (A) and (D) were performed at the same time and the same panels for the control conditions are presented.

- *in Figure 8B, the error bars remain to be defined.*

Response

The error bars are now defined in the Figure legend.

- *Please add a header/legend in Table EV1, within the file itself.*

Response

A legend is now provided within the Table EV1 file.

- *Please carefully double-check the reference list to make sure all citations are complete (some are currently missing page numbers/locators, e.g., Sung et al 2016).*

Response

Citations are now complete.

- *Please rename the Conflict of Interest section into "Disclosure and Competing Interests Statement", in accordance with our updated Guide to Authors (<https://www.embopress.org/competing-interests>)*

Response

The requested statement is now provided.

- *As we are switching from a free-text author contribution statement towards a more formal statement based on Contributor Role Taxonomy (CRediT) terms, please remove the present Author Contribution section and instead specify each author's contribution(s) directly in the Author Information page of our submission system during upload of the final manuscript. See <https://casrai.org/credit/> for more information.*

Response

The author's contribution section is now removed from the text and the required information is now specified in the author information page in the submission system.

- *Please adjust the order of the manuscript sections: Title page with complete author information, Abstract, Keywords, Introduction, Results, Discussion, Methods, Data*

Availability, Acknowledgements, Disclosure and Competing Interests Statement, References, Figure Legends.

Response

The manuscript order is now adjusted.

- Please include suggestions for a short 'blurb' text prefacing and summing up the conceptual aspect of the study in two sentences (max. 250 characters), followed by 3-5 one-sentence 'bullet points' with brief factual statements of key results of the paper; they will form the basis of an editor-written 'Synopsis' accompanying the online version of the article. Please also upload a synopsis image, which can be used as a "visual title" for the synopsis section of your paper. The image should be in PNG or JPG format, and please make sure that it remains in the modest dimensions of (exactly) 550 pixels wide and 300-600 pixels high.

Response

We now provide a “blurb” text followed by “bullet point” statements. We also provide a synopsis image.

- Finally, our routine text similarity checks found several passages in the Methods section being near-verbatim copies from previous papers by yourselves or others (e.g., 3D-RIM, CLEM, MS analysis). While this is in principle acceptable in the interest of reproducibility, the respective earlier articles should however be referenced at the start of each copied passage (e.g., "xxx was done essentially as reported by yyy et al").

Response

We have now rephrased many parts and cited the respective articles where the methods are also described.

Referee #1:

I believe that the additional text to put these results into a wider context in the introduction and the discussion is helpful. However, I think that even more background information would be helpful, and acknowledging the current state is not diminishing the quality of these results, e.g. I believe that the suggestion of the authors that "the role of the Ubiquitin system in aggregate formation/elimination ... remains poorly defined", could use some additional discussion/references to make clear what is already know, e.g. the presence of an autophagy-independent mechanism of aggregate clearance in the nucleus that relies on Hsp70, ubiquitination and the 26S proteasome has already been reported previously (Hjerpe et al Cell 2016).

Response

We have now expanded the introduction and discussion to incorporate the proposed changes. Specifically, we now include a detailed discussion of the findings by Hjerpe et al., 2016, which report an autophagy-independent mechanism of aggregate clearance in the nucleus involving Hsp70, Ubiquitination, and the 26S proteasome. We now clarify how our study builds upon this knowledge, emphasizing the novel aspects of our findings, such as that it is still unclear whether a PQC system operates within the nucleus for the elimination of stress-induced aggregates. We hope that we now highlight the advances and the contributions of our study in the field.

The authors convincingly show in their mass spec analysis of enriched nuclear fractions that other ribosomal proteins also follow the fate of RPL11. However I would have liked to see a somewhat more detailed discussion of the identified proteins, and the characterization of the two fractions analyzed. At the moment this experiment is not well connected to the rest of the manuscript, and it is very hard to understand the data in the supplemental table.

Response

We now provide additional details on the proteomics data, including a new table (Fig. 3) that highlights the most abundant groups of proteins identified within the isolated inclusions. This aims to facilitate the understanding of the key findings from this experiment. Additionally, we have expanded our description of the characterization of the analyzed fractions to better connect this experiment with the rest of the manuscript. However, as the primary goal of this experiment was to validate our imaging analysis of ribosomal proteins, we have chosen not to perform an extensive bioinformatic analysis of the proteomics data, to maintain the focus on our core findings. Finally, we have now simplified the supplemental table to present the proteomics data more clearly, explicitly indicating the groups of proteins quantified during MG132-induced stress and the recovery period.

It however is still not clear to me how it can be concluded from these experiments that the described quality control pathway is specific for nuclear protein quality control. This is especially important since there are experiments described in a manuscript deposited in BioRxiv that suggest that HUWE1 plays a general role in the clearance of aggregated proteins and that its role is not limited to aggregates in the nucleus (doi: <https://doi.org/10.1101/2023.05.30.542866>). As far as I can tell none of the experiments described there are contradicting the results presented here for RPL11, but they should influence the interpretation of the results, since a role of HUWE1 in the clearance of aggregates in the cytosol would suggest that this is not a quality control mechanism specific for the nucleus.

I believe these points can be addressed by augmenting the text.

Response

Thank you for highlighting the findings from the BioRxiv article on the role of HUWE1 in the cytoplasmic PQC. We now clarify our conclusions based on these reported roles of HUWE1.

We would also like to clarify that there are two separate aspects regarding the specificity of the proposed nuclear protein quality control (PQC) system:

1. Demonstration of a Nuclear PQC System:

Our study provides strong evidence for the existence of a PQC system that operates within the nucleus to eliminate nuclear inclusions. This evidence derives from:

- The photoconversion experiments.
- Results from autophagy and nuclear export inhibition.
- Fractionation experiments (Fig. 3) indicating that ribosomal protein elimination occurs exclusively in the nuclear compartment during the recovery period, with no similar process observed in the cytoplasm.

-The recruitment of nuclear proteasomes to inclusions, and the identification of nucleolar RNA Pol I-dependent lncRNAs (e.g. IGS42) necessary for nuclear inclusion elimination, further solidify this novel finding.

Collectively, these data support our conclusion that a PQC mechanism operates in the nucleus—a novel and significant aspect of our study.

2. **Broader Roles of PQC Components:**

While our findings highlight the nuclear specificity of the PQC system for the elimination of nuclear inclusions, we acknowledge that certain components of this system, such as HUWE1, HSP70, Ubiquitin and the proteasome, are also involved in cytoplasmic PQC processes. Therefore, we do not claim that all elements of the nuclear PQC system are exclusive to the nucleus.

We have now cited the BioRxiv study (DOI: <https://doi.org/10.1101/2023.05.30.542866>) in the Discussion section. We have also expanded our discussion to contextualize our findings in light of this study, stating that while HUWE1 has a role in the clearance of cytoplasmic aggregates, our results demonstrate its additional, distinct function in nuclear PQC. We hope that we now accurately describe our findings in the light of the BioRxiv study.

Minor points:

In the point-by-point response the authors state that: "Additionally, HUWE1 is shown both by proteomics and by imaging analysis to accumulate in nuclear inclusions upon stress." Can the authors point me towards the figure where the presence of HUWE1 is shown by "imaging analysis"?

Response

The proteomics and imaging analysis on HUWE1 is shown in the study by Maghames et al. 2018, Nat. Commun., Fig. 7. We cite this article.

In addition, our own proteomics data further confirm the presence of HUWE1 in stress-induced inclusions, and we discuss this finding in the context of our analysis.

The source data files seem to not always be annotated correctly e.g. the blots in Figure 3J seem to be in the folder for 3A.

Response

This has now been corrected as folder for Fig. 3 where raw data for all western blots are included.

Due to the complication of a non-peer reviewed finding that contradicts the main message of the manuscript as stated in the title, I feel not confident in judging the general significance and priority of this manuscript as well as the question if the conclusions of the paper are justified, and have decided to answer them with in the current version of the paper with "n/a". addressed by

Response

We do not agree that the findings of the BioRxiv paper contradict the main finding of our manuscript. The main finding of our study is the identification of a nuclear PQC pathway and not that HUWE1 operates exclusively in the nucleus. As discussed above these are 2 different aspects. The same argument holds for Ubiquitin, Proteasome and HSP70 that do not

exclusively operate in the nucleus. We have now revised the Discussion section to explicitly distinguish between the identification of a nuclear PQC pathway and the broader roles of its individual components, including HUWE1.

Referee #2:

I want to thank the authors for responding to the comments and for their extensive experimental work to revise the manuscript.

The authors stressed in the manuscript the significance of the recovery process of stress-induced nuclear inclusions, which requires the RNA Pol I-dependent production of the lncRNA IGS42. They claimed that lncRNA IGS42 localizes specifically within the formed inclusions and promotes their elimination by preserving the mobility of resident proteins. This important fact must be appropriately documented in the high-quality RNA FISH localization. I do not see it in Figure 9; the background is so high. The authors must significantly improve the quality of the RNA FISH detection of IGS42.

Response

We have now performed RNA FISH analysis for IGS42 using two different high resolution microscopy approaches (Fig. 9 and Appendix Fig. S8):

1. Airyscan confocal microscopy and include a 3D reconstruction.
2. RIM analysis.

We believe that both methods show a clear enrichment of the IGS42 RNA within the formed inclusions specifically upon stress. We want to emphasize that we do not claim that IGS42 is only localized within the formed inclusions, as indeed we see a weak localisation in the nuclear/nucleolar area. However, quantitation of the mean intensity shows a clear enrichment of IGS42 within the formed inclusions. A similar behavior is also observed for RPL11 where in addition to the formed inclusions RPL11 is also found within the nuclear/nucleolar area.

Referee #3:

While I was (and remain) excited about the data presented in this study, I am somewhat puzzled by the response of the authors to my request to test if the observed effect is specific for ISG42 or also depends on other stress-induced, pol I-dependent lncRNAs. In my request, I even listed three lncRNAs (IGS24, IGS27.5, IGS28) that from their own analysis seem to match their two selection criteria: stress induction and pol I dependency. For reasons that I do not comprehend, the authors performed the experiment instead for three lncRNAs (IGS16, 18, 20) that did not match their original selection criteria. Since these lncRNAs are not stress inducible and/or independent of pol I, it seems beforehand unlikely that they will have the same effect on clearance of inclusions as ISG42, which was indeed the outcome of the experiments. With that the question remains if the ability to make nuclear inclusion amendable to nuclear clearance mechanisms is unique to ISG42 or also applies to other stress-inducible, pol I-dependent lncRNAs. The outcome of the experiments may not affect the novelty or impact of the study but is relevant information for researchers who want to study the mechanisms underlying this phenomenon in more depth. If this experiment cannot be performed for technical reasons, this should be mentioned and the authors should discuss the possibility that they may be dealing with a more general feature of this class of lncRNAs in contrast to a unique property for ISG42.

Response

We apologize for the misunderstanding but in the 1st round of revision we did not understand why the reviewer proposed to test the effect of knockdown of IGS24, IGS27.5, IGS28, as none of them passed the significance t-test analysis, but indeed displayed qualitatively a similar response to IGS42. We thus opted to test lncRNAs that they have been proposed to play a role in nuclear protein inclusion dynamics for other stress conditions (heat shock, acidosis). We believe that this provides evidence for a stress-dependent role for such lncRNAs in inclusion dynamics, e.g. heat shock-acidosis vs proteasome inhibition. We now performed knockdown experiments for the lncRNAs proposed by the reviewer. However, we found that only IGS28 could be tested as the knockdown (using different siRNAs) of IGS24 and IGS27.5 caused extensive cell toxicity even in unstressed conditions. Similarly to what we observed for the knockdown of the other IGS lncRNAs, IGS28 knockdown had no effect on inclusion dynamics (Appendix Fig. S7). We want to emphasize that we do not claim that IGS42 is the only lncRNA that controls the elimination process. It only provides an example on how this class of lncRNAs induced upon stress can regulate inclusion dynamics. We now clearly describe all the above-mentioned details in the manuscript. This is also illustrated in the provided synopsis image, where we indicate generally the role of IGS lncRNAs in inclusion dynamics. We anticipate these data will initiate an interest within the scientific community for further detailed elucidation of the role of such lncRNAs or other type of RNAs in the nuclear PQC system.

Dr. Dimitris Xirodimas
CNRS
Centre de Recherche de Biochimie Macromoléculaire
Montpellier 34000
France

26th Nov 2024

Re: EMBOJ-2024-116906R1
A nuclear protein quality control system for elimination of nucleolus-related inclusions

Dear Dr. Xirodimas,

Thank you for submitting your re-revised manuscript for our consideration. I have now carefully assessed your responses and revisions to the last round of comments, and I am pleased to inform you that we have now accepted the paper for publication in The EMBO Journal.

Yours sincerely,

Hartmut Vodermaier
